# Towards Disentangled Preference Optimization Dynamics: Suppress the Loser, Preserve the Winner

**Wei Chen** [* 1]  **Yubing Wu** [* 1]  **Junmei Yang** [1]  **Delu Zeng** [1]  **Qibin Zhao** [2]  **John Paisley** [3]  **Min Chen** [1]  **Zhou Wang** [4]

## Abstract

Preference optimization is widely used to align large language models (LLMs) with human preferences. However, many margin-based methods also suppress the chosen response when they try to suppress the rejected one, and there is no general way to prevent this across different objectives. We address this issue with a unified *incentive-score decomposition* of preference optimization, revealing that different objectives share the same local update directions and differ only in their scalar weights. This decomposition provides a common framework for analyzing objectives that were previously studied in separate settings. Building on this decomposition, by analyzing the dynamics of the chosen/rejected likelihoods, we identify the *disentanglement band* (DB), a simple, testable condition that tells us when training can follow the desired path: suppress the loser while preserving the winner, possibly after an early stage. Using the DB, we propose *reward calibration* (RC), a plug-and-play method that adaptively rebalances the updates for chosen and rejected likelihoods to satisfy the DB, without redesigning the base objective. Empirical results show that RC leads to more disentangled dynamics, with better downstream performance observed across several settings. Our code is available at https://github.com/IceyWuu/DisentangledPreferenceOptimization.

## 1. Introduction

Aligning large language models (LLMs) with human preferences has shifted from online reinforcement learning from human feedback (RLHF) pipelines (Christiano et al., 2017; Bai et al., 2022; Ouyang et al., 2022) to stable offline optimization. Direct preference optimization (DPO) (Rafailov et al., 2023) catalyzed this shift by reframing alignment as a classification problem (Xiao et al., 2025), spawning a wide family of objectives, e.g., improving efficiency (Tang et al., 2024; Wang et al., 2024a), mitigating length bias (Park et al., 2024; Meng et al., 2024), and incorporating constraints (Xu et al., 2024a; Azar et al., 2024).

However, increasing the preference margin alone does not determine how the chosen and rejected likelihoods evolve. Ideally, training suppresses the rejected response without degrading the chosen one (Yuan et al., 2025), which we formalize as Pathway (iii) (Sec. 3.1). In contrast, many margin-based objectives, like DPO, exhibit entangled gradients (Yuan et al., 2025), where suppressing the loser also drags down the winner, causing likelihood displacement (Razin et al., 2025), the squeezing effect (Ren & Sutherland, 2025), and 3D-properties (Yan et al., 2025).

Some recent objectives treat the chosen and rejected terms more asymmetrically, often motivated by density ratio estimation (Sugiyama et al., 2012). In our diagnostics (e.g., Fig. 2a), objectives such as DIL (Xiao et al., 2025), DDRO (Higuchi & Suzuki, 2025), and BPO (Kim et al., 2025) more often avoid the lockstep winner–loser drift seen in margin-based objectives. We refer to this regime as *disentangled* dynamics (vs. *entangled* dynamics of margin-based objectives). Yet these objectives are often discussed as distinct families, leaving a broader question open:

*What objective property governs the induced likelihood dynamics, and how can we steer training toward Pathway (iii) even for entangled objectives?*

We address this with an *incentive-score decomposition*. Regardless of objective form, the gradient update decomposes along the same score directions; objective design mainly enters through two scalar incentives that weight them (Tab. 1). This yields a continuous-time analysis (Theorem 3.1) and a simple, testable condition—the ***disentanglement band*** (DB)—that characterizes when training can enter Pathway (iii) (Tab. 2), possibly after an early stage.

Finally, we translate this condition into a plug-and-play intervention. We propose *reward calibration* (RC), which adaptively rebalances chosen vs. rejected updates to make

---

[*]Equal contribution  [1]South China University of Technology  [2]AIP, RIKEN  [3]Columbia University  [4]University of Waterloo. Correspondence to: Delu Zeng <dlzeng@scut.edu.cn>.

*Proceedings of the $43^{rd}$ International Conference on Machine Learning*, Seoul, South Korea. PMLR 306, 2026. Copyright 2026 by the author(s).

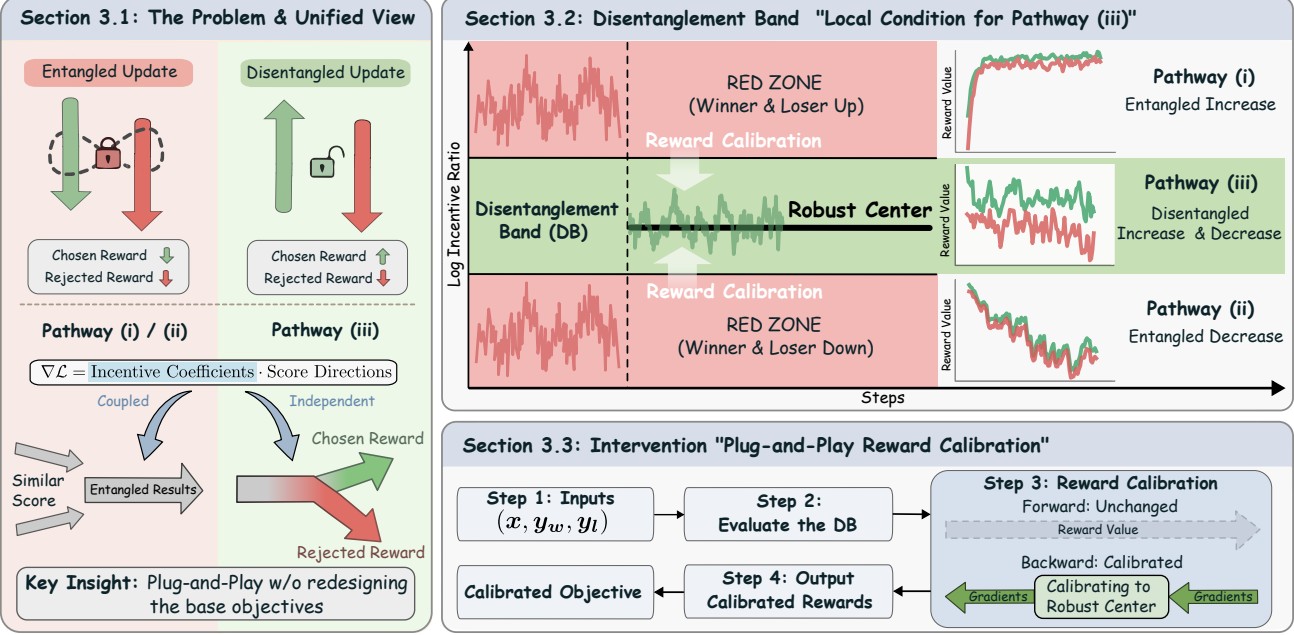

*Figure 1.* Overview of diagnosis and intervention for disentangled preference optimization dynamics. **Left:** Entangled objectives can couple winner and loser updates, leading to Pathway (i) (both likelihoods increase) or Pathway (ii) (both decrease). **Upper Right:** The disentanglement band (DB) provides a local, testable condition for entering Pathway (iii) (suppress the loser, preserve the winner). **Lower Right:** Reward calibration (RC) is a plug-and-play intervention that rescales backward gradients without changing the forward pass, making training more likely to fall into the DB and exhibit Pathway (iii), without redesigning the base objective.

Pathway (iii) more likely, without redesigning the base objective (overview in Fig. 1). RC often steers training toward the Pathway (iii) regime across objectives and model scales (see Fig. 4 and Sec. C.5), with gains that transfer to downstream benchmarks (see Tab. 4 and Sec. C.4).

To summarize, our contributions are:

1. We introduce an incentive-score decomposition that unifies entangled and disentangled objectives, isolating scalar incentives as key controls (Tab. 1).
2. We derive likelihood dynamics and identify the DB, a simple condition for the desired Pathway (iii): suppress the loser while preserve the winner.
3. We propose plug-and-play RC, which rebalances chosen vs. rejected updates to promote Pathway (iii) without redesigning the base objective.
4. We verify that RC improves likelihood dynamics across objectives and scales, with downstream improvements in several settings (Tab. 4, Sec. C.4).

## 2. Related Works and Background

### 2.1. Related Works

**LLM Alignment: From PPO to a Zoo of Objectives.**
For a long time, aligning LLMs with human preferences meant relying on the standard RLHF pipeline: train a reward model, then optimize the policy using PPO (Christiano et al., 2017; Schulman et al., 2017; Bai et al., 2022; Ouyang

et al., 2022). In practice, however, getting PPO to work reliably is tough. It is computationally expensive and notoriously sensitive to hyperparameters (Engstrom et al., 2020; Zheng et al., 2023). Consequently, the field shifted toward offline methods that skip the explicit reward modeling step. DPO (Rafailov et al., 2023) really kicked this door open, framing alignment as a stable classification problem.

Since then, numerous DPO variants have been proposed. Some address specific issues such as length bias (e.g., SimPO (Meng et al., 2024), R-DPO (Park et al., 2024)), while others introduce constraints to limit model drift (e.g., CPO (Xu et al., 2024a), IPO (Azar et al., 2024)). Additional work targets better sample efficiency (Tang et al., 2024; Wang et al., 2024a), robustness against noisy data (Chowdhury et al., 2024) and diverse LLM architectures and efficiency techniques (Li et al., 2024c; 2025c; 2026). Yet, while we have plenty of new objectives, we often lack a clear and unified picture of what they are actually doing to the model's internal probability updates.

**Entangled vs. Disentangled Objectives.** Broadly, preference optimization objectives fall into two categories. The first, and currently dominant, class relies on an entangled margin derived from the Bradley–Terry assumption (Bradley & Terry, 1952), as exemplified by DPO (Rafailov et al., 2023), IPO (Azar et al., 2024), and CPO (Xu et al., 2024a). By coupling the chosen and rejected likelihoods through

a single margin, these objectives limit asymmetric control (Yuan et al., 2025). As a result, suppressing the rejected likelihood often also depresses the chosen one, leading to phenomena such as the likelihood displacement (Razin et al., 2025) and gradient entanglement (Yuan et al., 2025).

This limitation has motivated a newer line of work explicitly decouples winner and loser rewards. Such disentanglement appears in methods for unpaired data, such as KTO (Ethayarajh et al., 2024), as well as in density ratio-based objectives. For example, DIL (Xiao et al., 2025) is derived from imitation learning bounds, while DDRO (Higuchi & Suzuki, 2025) and BPO (Kim et al., 2025) build on statistical consistency and Bregman divergences (Bregman, 1967; Sugiyama et al., 2012). By disentangling chosen and rejected rewards, these methods enable asymmetric control over updates, often yielding more stable and selective suppression of rejected responses.

**Interpretability of Preference Optimization and Learning Dynamics.** Proposing a new objective is one thing; explaining why it succeeds or fails during optimization is another. Recent work has increasingly sought to interpret preference optimization by examining its assumptions and learning dynamics. Some studies question the rigidity of the Bradley–Terry assumption (Azar et al., 2024; Munos et al., 2024; Ethayarajh et al., 2024) or analyze the impact of noisy preference labels (Chowdhury et al., 2024), while others focus on downstream pathologies such as length bias exploitation (Park et al., 2024), reward over-optimization (Gao et al., 2023), and likelihood displacement relative to the reference model (Razin et al., 2025; Pal et al., 2024).

Most closely related are dynamical analyses of preference optimization. Ren & Sutherland (2025) study likelihood shifts during fine-tuning, while Yuan et al. (2025) identify gradient entanglement in margin-based objectives. Although continuous-time dynamics have been used more broadly (Li et al., 2024a; 2025a; 2024b), existing work remains fragmented: margin-based objectives are analyzed dynamically, whereas density ratio methods (e.g., BPO (Kim et al., 2025), DIL (Xiao et al., 2025), DDRO (Higuchi & Suzuki, 2025)) are not. We bridge this gap by unifying entangled and disentangled objectives under a common dynamical framework via incentive-score decomposition.

### 2.2. Background

**Notations.** Each datapoint is a triplet $(\boldsymbol{x}, \boldsymbol{y}_w, \boldsymbol{y}_l)$ with prompt $\boldsymbol{x}$ and responses $\boldsymbol{y}_w$ ("winner") and $\boldsymbol{y}_l$ ("loser"). Let $\pi_{\boldsymbol{\theta}}$ be the policy parameterized by $\boldsymbol{\theta}$, and $\pi_{\mathrm{ref}}$ an optional fixed reference policy. We define the sequence-level log-probabilities $z_w(\boldsymbol{\theta}) \triangleq \log \pi_{\boldsymbol{\theta}}(\boldsymbol{y}_w \mid \boldsymbol{x})$ and $z_l(\boldsymbol{\theta}) \triangleq \log \pi_{\boldsymbol{\theta}}(\boldsymbol{y}_l \mid \boldsymbol{x})$, the preference margin $m(\boldsymbol{\theta}) \triangleq z_w(\boldsymbol{\theta}) - z_l(\boldsymbol{\theta})$, and the parameter scores $\boldsymbol{s}_w(\boldsymbol{\theta}) \triangleq \nabla_{\boldsymbol{\theta}} z_w(\boldsymbol{\theta})$, $\boldsymbol{s}_l(\boldsymbol{\theta}) \triangleq \nabla_{\boldsymbol{\theta}} z_l(\boldsymbol{\theta})$. When a reference policy is used, we further define the reference-normalized log-probabilities

$\tilde{z}_w(\boldsymbol{\theta}) \triangleq z_w(\boldsymbol{\theta}) - \log \pi_{\mathrm{ref}}(\boldsymbol{y}_w \mid \boldsymbol{x})$ and $\tilde{z}_l(\boldsymbol{\theta}) \triangleq z_l(\boldsymbol{\theta}) - \log \pi_{\mathrm{ref}}(\boldsymbol{y}_l \mid \boldsymbol{x})$, and the effective margin $\tilde{m}(\boldsymbol{\theta}) \triangleq m(\boldsymbol{\theta}) - m_{\mathrm{ref}}$ where $m_{\mathrm{ref}} \triangleq \log \pi_{\mathrm{ref}}(\boldsymbol{y}_w \mid \boldsymbol{x}) - \log \pi_{\mathrm{ref}}(\boldsymbol{y}_l \mid \boldsymbol{x})$. All quantities are defined pointwise for a given triplet. See Tab. 6 in Sec. A.1 for a full summary.

**Entangled Margin-Based Objectives.** For a given triplet $(\boldsymbol{x}, \boldsymbol{y}_w, \boldsymbol{y}_l)$, a broad class of margin-based preference objectives adopt a *entangled* template (Yuan et al., 2025):
$$\mathcal{L}(z_w, z_l) = \ell\left(h_w(z_w) - h_l(z_l) - c\right) + \Lambda(z_w), \quad (1)$$

where $\ell(\cdot)$ is an outer shaping function, $h_w$ and $h_l$ are transformations, $c$ is a constant offset, and $\Lambda(z_w)$ is an optional chosen-only term, e.g., a likelihood regularizer in SlicHF (Zhao et al., 2023). Margin-based variants can be interpreted as instantiations of Eq. (1) by setting different $\ell$, $h_w$, $h_l$, $c$, and $\Lambda$ (see Sec. A.3 for details).

In this case, DPO (Rafailov et al., 2023) is an entangled objective with $\ell(\cdot) = -\log \sigma(\cdot), h_{w|l}(z_{w|l}) = \beta z_{w|l}$:

$$\mathcal{L}_{\mathrm{DPO}}(z_w, z_l) = -\log \sigma\left(\beta\left(z_w - z_l - z_w^{\mathrm{ref}} + z_l^{\mathrm{ref}}\right)\right), \quad (2)$$

where $\beta > 0$ is a hyperparameter, $z_w^{\mathrm{ref}}$ and $z_l^{\mathrm{ref}}$ are constants for a given reference model $\pi_{\mathrm{ref}}$ and triplet $(\boldsymbol{x}, \boldsymbol{y}_w, \boldsymbol{y}_l)$.

## 3. Disentanglement Band and Plug-and-Play Reward Calibration

In this section, we analyze preference optimization through training-time likelihood dynamics. Motivated by empirical differences between entangled and disentangled objectives, we study how objective structure shapes likelihood trajectories and enables the desired behavior, and how this analysis yields a simple, practical way to steer these dynamics.

### 3.1. Unifying Objectives via Incentives

**The Three Pathways of Margin Growth.** Under the Bradley-Terry assumption, most alignment objectives aim to increase the preference margin $m = z_w - z_l$. However, a larger margin is a scalar outcome that can be achieved via three qualitatively distinct pathways in the $(z_w, z_l)$ plane:

- (i) *Entangled increase*: both increase, with $z_w$ faster;
- (ii) *Entangled decrease*: both decrease, with $z_w$ slower;
- (iii) *Disentangled updates*: $z_w$ is not driven downward (and may increase) while $z_l$ is suppressed.

**Pathway (iii)** is a natural desideratum in the post-SFT regime (Yuan et al., 2025), as it avoids likelihood displacement and gradient entanglement without degrading the chosen response. Although not universally optimal, it provides a principled direction for training dynamics.

However, many margin-based objectives entangle the updates of the chosen and rejected rewards, preventing them from changing independently. This gradient entanglement (Yuan et al., 2025) causes pressure to decrease $z_l$ often

also decreases $z_w$, yielding the Pathway (ii) (see Fig. 2a for illustration). Such objectives fit the entangled template in Eq. (1), with different settings (see Sec. A.3 for details):

$$\mathcal{L}(z_w, z_l) = \ell\left(h_w(z_w) - h_l(z_l) - c\right) + \Lambda(z_w). \quad (3)$$

Recently, a growing class of disentangled, density ratio-based objectives (Xiao et al., 2025; Higuchi & Suzuki, 2025) has been observed to often exhibit more disentangled reward dynamics than margin-based objectives, frequently suppressing $z_l$ without strongly degrading $z_w$ in many settings (see Fig. 2a for a comparison of DPO and DIL-BCE (Xiao et al., 2025)). These objectives are disentangled in the sense that they act on two separate scalars,

$$\mathcal{L}(z_w, z_l) = \ell_w(z_w - z_w^{\text{ref}}) + \ell_l(z_l - z_l^{\text{ref}}), \quad (4)$$

where $\ell_w$ and $\ell_l$ are outer shaping functions corresponding to $z_w$ and $z_l$, respectively. This enables more flexible and potentially asymmetric control over maintaining $z_w$ versus suppressing $z_l$, i.e. Pathway (iii). We summarize this contrast with a schematic comparison in Fig. 2.

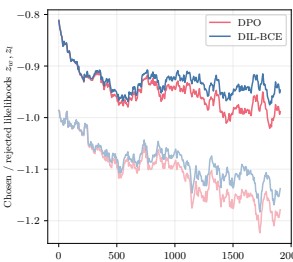 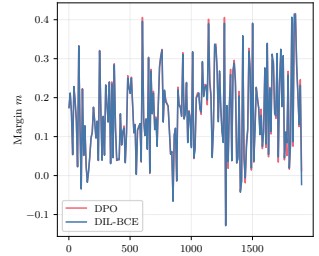

*(a)* Likelihoods over steps.    *(b)* Margin over steps.

*Figure 2.* **(a)** Likelihoods over steps under entangled (DPO (Rafailov et al., 2023)) and disentangled (DIL-BCE (Xiao et al., 2025)) objectives. **(b)** Both increase the margin during training, but induce different pathways. Mistral-7B is used here.

However, the underlying mechanism remains unclear. These observations raise a key question: *which objective properties determine when training enters the desired Pathway (iii)?*

**Decomposing the Gradient: Incentives and Scores.** To explain these differences, we look at the gradients. Regardless of whether the objective is entangled (Eq. (3)) or disentangled (Eq. (4)), the gradient with respect to parameters $\boldsymbol{\theta}$ always decomposes along the same two score directions $\boldsymbol{s}_w$ and $\boldsymbol{s}_l$. Formally, for any objective $\mathcal{L}(z_w, z_l)$, the chain rule yields a unified decomposition of the negative gradient

$$-\nabla_{\boldsymbol{\theta}}\mathcal{L}(z_w(\boldsymbol{\theta}), z_l(\boldsymbol{\theta})) = -\frac{\partial\mathcal{L}}{\partial z_w}\nabla_{\boldsymbol{\theta}}z_w(\boldsymbol{\theta}) - \frac{\partial\mathcal{L}}{\partial z_l}\nabla_{\boldsymbol{\theta}}z_l(\boldsymbol{\theta})$$
$$\triangleq d_w(\boldsymbol{\theta})\boldsymbol{s}_w(\boldsymbol{\theta}) - d_l(\boldsymbol{\theta})\boldsymbol{s}_l(\boldsymbol{\theta}), \quad (5)$$

where $d_w(\boldsymbol{\theta}) \triangleq -\frac{\partial\mathcal{L}}{\partial z_w}, d_l(\boldsymbol{\theta}) \triangleq \frac{\partial\mathcal{L}}{\partial z_l}$ are the *incentive coefficients*. Eq. (5) reveals a crucial insight: the update directions $(\boldsymbol{s}_w, \boldsymbol{s}_l)$ are largely fixed by the model and data (via $\boldsymbol{s}_w, \boldsymbol{s}_l$), while the objective design is entirely distilled into the scalars $(d_w, d_l)$. These coefficients act as the knobs that

control the magnitude and balance of the update. Detailed derivations of $(d_w, d_l)$ for common objectives are provided in Sec. A.3 and summarized in Tab. 1. In the next subsection, we connect the unified decomposition in Eq. (5) to a continuous-time dynamical system for $(z_w, z_l)$, which yields an explicit condition for realizing Pathway (iii).

*Table 1.* Incentive coefficients $(d_w, d_l)$ for preference optimization objectives (notations follow Sec. 2.2). Hyperparameters include $\beta, \lambda, \gamma, \alpha$ and $\sigma$ denotes the sigmoid. See Sec. A.3 for derivations.

| Method | Chosen incentive $d_w$ | Rejected incentive $d_l$ | $d_w = d_l$ |
|--------|------------------------|--------------------------|-------------|
| *Entangled Objectives* | | | |
| DPO | $\beta\sigma(-\beta\tilde{m})$ | $\beta\sigma(-\beta\tilde{m})$ | ✓ |
| TI-DPO | $\beta\sigma(-\beta\tilde{m})$ | $\beta\sigma(-\beta\tilde{m})$ | ✓ |
| IPO | $2(\lambda - \tilde{m})$ | $2(\lambda - \tilde{m})$ | ✓ |
| R-DPO | $\beta\sigma(-\beta\tilde{m} - \alpha\Delta|\boldsymbol{y}|)$ | $\beta\sigma(-\beta\tilde{m} - \alpha\Delta|\boldsymbol{y}|)$ | ✓ |
| RRHF | $\mathbb{I}(m < 0) + \lambda$ | $\mathbb{I}(m < 0)$ | ✗ |
| SlicHF | $\mathbb{I}(m < \gamma) + \lambda$ | $\mathbb{I}(m < \gamma)$ | ✗ |
| SimPO | $\frac{\beta}{\|\boldsymbol{y}_w\|}\sigma(\gamma - m_{\text{norm}})$ | $\frac{\beta}{\|\boldsymbol{y}_l\|}\sigma(\gamma - m_{\text{norm}})$ | ✗ |
| CPO | $\beta\sigma(-\beta m) + \lambda$ | $\beta\sigma(-\beta m)$ | ✗ |
| *Disentangled Objectives* | | | |
| KTO | $\lambda_w\sigma'(-\tilde{z}_w)$ | $\lambda_l\sigma'(\tilde{z}_l)$ | ✗ |
| DDRO | $1$ | $\frac{\exp(\tilde{z}_l)}{2 - \exp(\tilde{z}_l)}$ | ✗ |
| DIL-BCE | $\sigma(-\tilde{z}_w)$ | $\sigma(\tilde{z}_l)$ | ✗ |
| DIL-UKL | $1$ | $\exp(\tilde{z}_l)$ | ✗ |
| DIL-LSIF | $\exp(\tilde{z}_w)$ | $\exp(2\tilde{z}_l)$ | ✗ |

**Case Studies: Entangled vs. Disentangled Incentives.** The difference between entangled and disentangled objectives is reflected in their incentive structures (see Tab. 1). Entangled objectives like DPO and IPO have tightly coupled incentives, often symmetric ($d_w = d_l$) or constrained by the shared margin $m$, so that increasing the winner inherently affects the loser. Disentangled objectives like DIL-BCE, by contrast, assign independent incentives: $d_w = \sigma(-\tilde{z}_w)$ depends only on the chosen response, while $d_l = \sigma(\tilde{z}_l)$ depends only on the rejected one, thereby avoiding the entangled dynamics of margin-based objectives.

### 3.2. Likelihood Dynamics and Pathway Conditions

By expressing an objective via its incentive coefficients $(d_w, d_l)$, likelihood analysis reduces to these scalars. Viewing gradient descent as a continuous flow then yields conditions on $(d_w, d_l)$ for Pathway (iii), revealing structural requirements of the base objective.

**Theorem 3.1.** *Consider a parameter trajectory $\boldsymbol{\theta}_t$ driven by the gradient flow of Eq. (5). Along $\boldsymbol{\theta}_t$, define $z_{w|l,t} \triangleq z_{w|l}(\boldsymbol{\theta}_t)$, $\boldsymbol{s}_{w|l,t} \triangleq \nabla_{\boldsymbol{\theta}}z_{w|l}(\boldsymbol{\theta}_t)$, and $d_{w|l,t} \triangleq d_{w|l}(\boldsymbol{\theta}_t)$, with $w|l$ being $w$ or $l$. Then, the continuous-time dynamics of the chosen and rejected likelihoods are governed by:*

$$\frac{\partial z_{w,t}}{\partial t} = \dot{z}_{w,t} = d_{w,t}\|\boldsymbol{s}_{w,t}\|^2 - d_{l,t}\langle\boldsymbol{s}_{w,t}, \boldsymbol{s}_{l,t}\rangle,$$
$$\frac{\partial z_{l,t}}{\partial t} = \dot{z}_{l,t} = d_{w,t}\langle\boldsymbol{s}_{w,t}, \boldsymbol{s}_{l,t}\rangle - d_{l,t}\|\boldsymbol{s}_{l,t}\|^2. \quad (6)$$

**Corollary 3.2.** *Define the time-indexed margin $m_t \triangleq z_{w,t} - z_{l,t}$. Then, the continuous-time dynamic of $m_t$ satisfy*

$$\dot{m}_t = d_{w,t}\|\boldsymbol{s}_{w,t}\|^2 + d_{l,t}\|\boldsymbol{s}_{l,t}\|^2 - (d_{w,t} + d_{l,t})\langle \boldsymbol{s}_{w,t}, \boldsymbol{s}_{l,t}\rangle.$$

See Sec. B.1 and Sec. B.2 for the detailed proofs of Theorem 3.1 and Corollary 3.2, respectively.

**Mechanistic Interpretation.** Theorem 3.1 provides a clear mechanistic interpretation of gradient entanglement, revealing that the dynamic of the chosen likelihood, $\dot{z}_{w,t}$, is governed by two opposing terms. The self-reinforcement term, $d_{w,t}\|\boldsymbol{s}_{w,t}\|^2$, pushes $z_{w,t}$ upward in proportion to its own incentive and score magnitude. In contrast, the coupling drag term, $-d_{l,t}\langle \boldsymbol{s}_{w,t}, \boldsymbol{s}_{l,t}\rangle$, pulls it downward due to interference from updates targeting the rejected response.

In practice, scores are typically positively correlated ($\langle \boldsymbol{s}_{w,t}, \boldsymbol{s}_{l,t}\rangle > 0$, as illustrated in Fig. 3a). If the objective assigns too much weight to the rejected incentive, e.g., $d_{l,t} = d_{w,t}$ as in DPO, the drag can dominate, causing $\dot{z}_{w,t} < 0$ even when the desired behavior is to avoid driving $z_{w,t}$ downward. This explains the squeezing effect (Ren & Sutherland, 2025). These dynamics are local-in-time, so training may enter the desired regime only after an early stage, consistent with our empirical results (Fig. 2a).

Corollary 3.2 shows that margin growth is driven by $(d_{w,t}, d_{l,t})$ but regulated by the coupling $\langle \boldsymbol{s}_{w,t}, \boldsymbol{s}_{l,t}\rangle$, which quantifies winner–loser interaction. We use Theorem 3.1 and Corollary 3.2 as analytical abstractions to expose objective-induced drift and derive sign conditions, rather than to predict exact trajectories.

**Conditions of the Three Pathways.** We now translate the qualitative Pathways (i)–(iii) in Sec. 3.1 into local sign conditions implied by the likelihood dynamics in Theorem 3.1. In particular, the Pathway (iii) regime corresponds to $\dot{z}_{w,t} \geq 0$ and $\dot{z}_{l,t} \leq 0$.

The key observation is that these signs depend on how strongly the objective updates the winner versus the loser relative to their score coupling. We therefore normalize the coupling term via the cosine similarity:

$$\rho_t \triangleq \frac{\langle \boldsymbol{s}_{w,t}, \boldsymbol{s}_{l,t}\rangle}{\|\boldsymbol{s}_{w,t}\|\|\boldsymbol{s}_{l,t}\|} \in [-1, 1]. \qquad (7)$$

We focus on the challenging regime where $\rho_t > 0$, which is typical in fine-tuning tasks (see Fig. 3a). For standard preference objectives, we typically have $d_{w,t} > 0$ and $d_{l,t} > 0$ (see Tab. 1). Substituting $\langle \boldsymbol{s}_{w,t}, \boldsymbol{s}_{l,t}\rangle = \rho_t\|\boldsymbol{s}_{w,t}\|\|\boldsymbol{s}_{l,t}\|$ into Eq. (6) in Theorem 3.1 and factoring out $d_{l,t}$ yields

$$\dot{z}_{w,t} = d_{l,t}\|\boldsymbol{s}_{w,t}\|^2 \left(\frac{d_{w,t}}{d_{l,t}} - \rho_t \frac{\|\boldsymbol{s}_{l,t}\|}{\|\boldsymbol{s}_{w,t}\|}\right),$$

$$\dot{z}_{l,t} = d_{l,t}\|\boldsymbol{s}_{l,t}\|^2 \left(\rho_t \frac{\|\boldsymbol{s}_{w,t}\|}{\|\boldsymbol{s}_{l,t}\|}\frac{d_{w,t}}{d_{l,t}} - 1\right). \qquad (8)$$

Hence the signs of $(\dot{z}_{w,t}, \dot{z}_{l,t})$ are governed by where the ratio $d_{w,t}/d_{l,t}$ falls relative to two thresholds, $\rho_t \frac{\|\boldsymbol{s}_{l,t}\|}{\|\boldsymbol{s}_{w,t}\|}$ and $\frac{1}{\rho_t}\frac{\|\boldsymbol{s}_{l,t}\|}{\|\boldsymbol{s}_{w,t}\|}$, which partition the ratio axis into three regimes that exactly match Pathways (i)-(iii), as summaried in Tab. 2.

*Table 2.* For $\rho_t > 0$, the band thresholds partition the ratio axis into three regimes that exactly match Pathways (i)-(iii) in Sec. 3.1.

| Regime on $d_{w,t}/d_{l,t}$ | $(\dot{z}_{w,t}, \dot{z}_{l,t})$ | Pathway |
|---|---|---|
| $\left(0, \rho_t \frac{\|\boldsymbol{s}_{l,t}\|}{\|\boldsymbol{s}_{w,t}\|}\right)$ | $(< 0, < 0)$ | (ii) both decrease |
| $\left[\rho_t \frac{\|\boldsymbol{s}_{l,t}\|}{\|\boldsymbol{s}_{w,t}\|}, \frac{1}{\rho_t}\frac{\|\boldsymbol{s}_{l,t}\|}{\|\boldsymbol{s}_{w,t}\|}\right]$ | $(\geq 0, \leq 0)$ | (iii) suppress $z_{l,t}$, preserve $z_{w,t}$ |
| $\left(\frac{1}{\rho_t}\frac{\|\boldsymbol{s}_{l,t}\|}{\|\boldsymbol{s}_{w,t}\|}, +\infty\right)$ | $(> 0, > 0)$ | (i) both increase |

**Disentanglement Band.** Only the second regime in Tab. 2 permits the Pathway (iii). Equivalently, the conditions of Pathway (iii), i.e., $(\dot{z}_{w,t}, \dot{z}_{l,t}) = (\geq 0, \leq 0)$ hold if

$$\log\frac{\|\boldsymbol{s}_{l,t}\|}{\|\boldsymbol{s}_{w,t}\|} + \log\rho_t \leq \log r_t \leq \log\frac{\|\boldsymbol{s}_{l,t}\|}{\|\boldsymbol{s}_{w,t}\|} - \log\rho_t, \quad (9)$$

where $r_t \triangleq d_{w,t}/d_{l,t}$ for simplicity. We term this interval the **disentanglement band (DB)**, since any $\log r_t$ within this band makes the sign pattern $(\dot{z}_{w,t}, \dot{z}_{l,t}) = (\geq 0, \leq 0)$ feasible under Eq. (8), i.e., it admits Pathway (iii) at time $t$.

**Interpreting the Band.** The DB provides a unified explanation for the behavior of different objectives (Tab. 2):

- *Inside DB*: The incentive ratio $r_t$ is balanced to overcome the correlation $\rho_t$. The update suppresses the loser and preserves the winner (Pathway (iii)).
- *Below DB* ($r_t < \rho_t \frac{\|\boldsymbol{s}_{l,t}\|}{\|\boldsymbol{s}_{w,t}\|}$): The chosen incentive $d_{w,t}$ is too weak relative to $d_{l,t}$. The "drag" from the loser update pulls $z_w$ down (Pathway (ii)).
- *Above DB* ($r_t > \frac{1}{\rho_t}\frac{\|\boldsymbol{s}_{l,t}\|}{\|\boldsymbol{s}_{w,t}\|}$): The chosen incentive is too strong, causing both $z_w$ and $z_l$ to rise (Pathway (i)).

Crucially, the DB width is $2|\log\rho_t|$. As $\rho_t \to 1$, the band collapses, leaving little room to satisfy $(\dot{z}_{w,t}, \dot{z}_{l,t}) = (\geq 0, \leq 0)$ under first-order analysis. Yet $\rho_t < 1$ in practice, yielding a feasible optimization window (Fig. 3a).

**When Disentanglement Succeeds?** Two factors determine disentanglement success: (1) $\rho_t$ sets the feasible DB width in Eq. (9), and (2) the objective determines where the realized log-ratio $\log r_t$ lands within DB.

Thus, *a wide DB alone is insufficient*: entanglement persists if $\log r_t$ consistently lies outside the band or clusters near its thresholds. This behavior is evident in Fig. 3. For instance, CPO achieves small $\rho_t$ (hence a wide DB) but often pushes $r_t$ to the upper edge or beyond (Fig. 3c), following Pathway (i) (Fig. 3b). In contrast, disentangled objectives like DIL-BCE keep $r_t$ stably inside the DB and closer to its center (Fig. 3d), aligning with Pathway (iii).

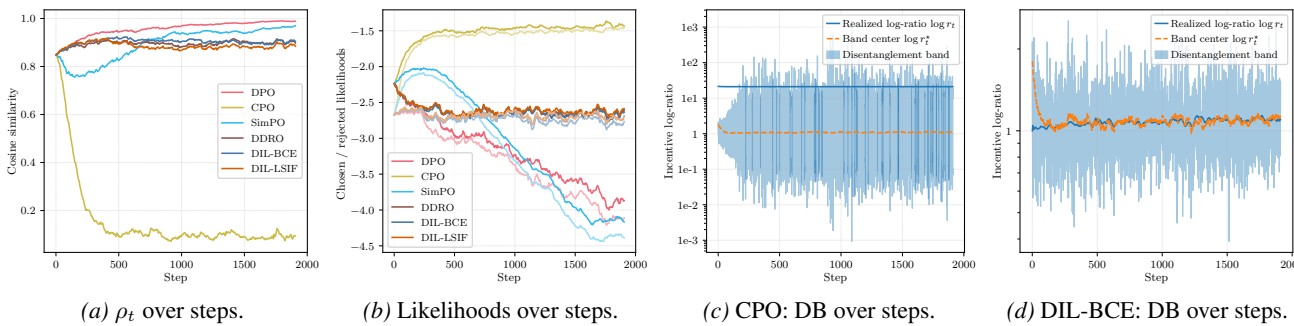

*(a) $\rho_t$ over steps.*     *(b)* Likelihoods over steps.     *(c)* CPO: DB over steps.     *(d)* DIL-BCE: DB over steps.

*Figure 3.* **(a)** Cosine similarity $\rho_t$ between score directions $\boldsymbol{s}_{w,t}$ and $\boldsymbol{s}_{l,t}$; larger $\rho_t$ indicates stronger winner-loser coupling and a narrower DB in Eq. (9). **(b)** Corresponding likelihood trajectories $(z_{w,t}, z_{l,t})$, indicating whether training reaches the Pathway (iii) regime (suppress $z_{l,t}$, preserve $z_{w,t}$), or follows entangled Pathways (i)/(ii). **(c)(d)** A wide DB alone is insufficient: even with a wide DB (small $\rho_t$), training can remain entangled when $\log r_t$ stays outside the band (CPO, Fig. 3c). In contrast, keeping $\log r_t$ inside the DB is closely associated with entering the Pathway (iii) regime in our experiments (DIL-BCE, Fig. 3d). See Sec. C.5 for full results across models (Pythia-410M, Pythia-2B, Mistral-7B, and Qwen2.5-7B) and objectives.

Moreover, from Eq. (8), the lower edge of the DB corresponds exactly to the locus where $\dot{z}_{w,t} = 0$. Slightly above this edge, the winner remains nearly stable while $\dot{z}_{l,t} < 0$ is still enforced. In practice, however, $r_t$ is not a deterministic knob: it fluctuates over steps and across triplets under minibatch sampling and adaptive optimization. Consequently, a robust objective should keep $r_t$ stably inside DB rather than merely avoiding crossing the lower edge. This motivates analyzing where $\log r_t$ sits within DB, beyond feasibility.

**Where the Log-Ratio Should Sit?** To maximize robustness against multiplicative fluctuations in the ratio, we seek the point that maximizes the log-distance to the nearest threshold. Let $\Delta_t(\log r_t)$ denote the minimum distance from $\log r_t$ to the lower and upper thresholds of the DB:

$$\Delta_t(\log r_t) \triangleq \min\left(\log \frac{r_t \|\boldsymbol{s}_{w,t}\|}{\rho_t \|\boldsymbol{s}_{l,t}\|}, \log \frac{\|\boldsymbol{s}_{l,t}\|}{\rho_t r_t \|\boldsymbol{s}_{w,t}\|}\right). \quad (10)$$

Then $\Delta_t(\log r_t) \geq 0$ if and only if $r_t$ lies inside DB, and larger $\Delta_t(\log r_t)$ means more tolerance to multiplicative fluctuations of $\log r_t$ while preserving pathway (iii).

**Proposition 3.3.** *For given $\rho_t$ and $\|\boldsymbol{s}_{l,t}\|/\|\boldsymbol{s}_{w,t}\|$, the maximizer of $\Delta_t(\log r_t)$ over all $\log r_t$ inside DB satisfies*

$$\log r_t^\star = \arg \max_{\log r_t \in DB} \Delta_t(\log r_t) = \log \frac{\|\boldsymbol{s}_{l,t}\|}{\|\boldsymbol{s}_{w,t}\|}. \quad (11)$$

See Sec. B.3 for a detailed proof. The robust center $\log r_t^\star$ balances the score vector norms. To maintain stability, incentives should be inversely proportional to the gradient magnitudes of the respective likelihoods.

### 3.3. Plug-and-Play Reward Calibration

A natural next question is: *how can we make the realized log-ratio $\log r_t$ track $\log r_t^\star$ during stochastic training?*

Based on this robust centering, we propose a simple wrapper method: **reward calibration (RC)**. Instead of redesigning the objective function $\mathcal{L}$ from scratch, we can adaptively rescale the likelihood gradients to make $\log r_t$ track $\log r_t^\star$.

We define the calibrated likelihoods as follows:

$$\begin{aligned}
z_{w,t}^{\mathrm{rc}} &\triangleq \left(\frac{r_t^\star}{r_t}\right)^{1/2} z_{w,t} + \left(1 - \left(\frac{r_t^\star}{r_t}\right)^{1/2}\right) \mathrm{sg}(z_{w,t}), \\
z_{l,t}^{\mathrm{rc}} &\triangleq \left(\frac{r_t^\star}{r_t}\right)^{-1/2} z_{l,t} + \left(1 - \left(\frac{r_t^\star}{r_t}\right)^{-1/2}\right) \mathrm{sg}(z_{l,t}),
\end{aligned} \quad (12)$$

where $\mathrm{sg}(\cdot)$ denotes a stop-gradient operator, satisfying $\mathrm{sg}(x) = x$ and $\frac{\partial}{\partial x} \mathrm{sg}(x) = 0$.

Applying the base objective $\mathcal{L}$ to the calibrated likelihoods leaves the forward pass unchanged ($z^{\mathrm{rc}} = z$), but scales the backward gradients by $(r_t^\star/r_t)^{1/2}$ and $(r_t^\star/r_t)^{-1/2}$, respectively. This adaptively calibrates the incentive coefficients $(d_{w,t}, d_{l,t})$ so that the incentive log-ratio aligns with the target center $\log r_t^\star$, rectifying the dynamics on the fly. The next proposition formalizes this effect.

**Proposition 3.4.** *The log-ratio induced by the calibrated likelihoods in Eq. (12) is exactly the log-ratio $\log r_t^\star$ defined in Proposition 3.3.*

See Sec. B.4 for a detailed proof.

Building on Proposition 3.4, we introduce a plug-and-play calibration wrapper applicable to a broad class of preference optimization methods. The procedure is summarized in Algorithm 1, with a PyTorch implementation provided in Algorithm 2. Importantly, the method requires no changes to the underlying optimizer and objective formulations; it merely rescales likelihood gradients via calibrated inputs $(z_{w,t}^{\mathrm{rc}}, z_{l,t}^{\mathrm{rc}})$ to promote the Pathway (iii) regime by keeping the incentive ratio in (and near the center of) the DB.

**Practical Implementations for Reward Calibration.** To ensure efficiency and numerical stability, we adopt two implementation choices. First, we approximate the scores $(\|\boldsymbol{s}_{w,t}\|, \|\boldsymbol{s}_{l,t}\|)$ using gradients from the output layer: the language model head in full fine-tuning, or the final adapter modules in LoRA (Hu et al., 2022) or QLoRA (Dettmers et al., 2023). Empirically, this avoids the cost of a full backward pass and yields similar DB values to full-parameter

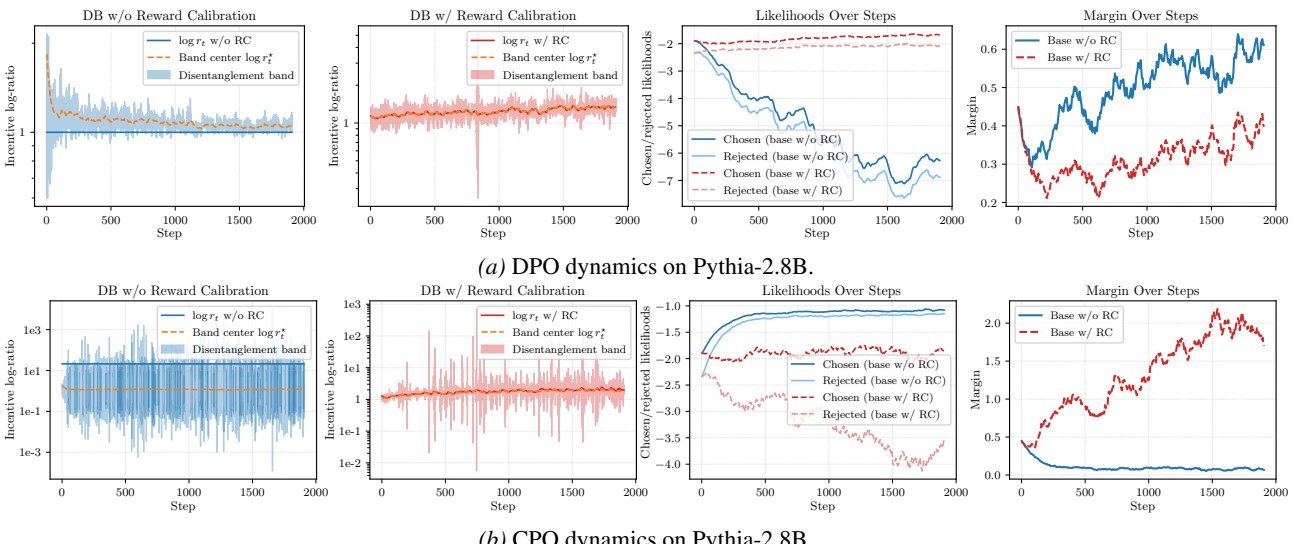

*(a)* DPO dynamics on Pythia-2.8B.

*(b)* CPO dynamics on Pythia-2.8B.

*Figure 4.* Impact of RC on preference dynamics on Pythia-2.8B. Each panel (DPO/CPO) reports four coupled trajectories over training steps: DB w/ or w/o RC, likelihood trajectories, and margin growth. Without RC, runs frequently drift toward DB boundaries or violate the band, which often coincides with undesired likelihood drifts. With RC, $\log r_t$ is pulled toward the DB center, making it more likely for training to enter and remain in the Pathway (iii) regime (typically after a short transient) while preserving margin improvement. See Sec. C.5 for full results across models (Pythia-410M, Pythia-2B and Mistral-7B) and objectives.

gradients (see Fig. 6 in Sec. C.3). Second, to reduce the high variance of stochastic gradients with batch size 1, we use log-domain exponential moving averages (EMA) to estimate the geometric mean of $\log r_t^\star$ and $\log r_t$ for calibration. Further details are in Sec. C.3.

---

**Algorithm 1** Plug-and-Play Reward Calibration

---

1: **Input:** Triplet $(\boldsymbol{x}, \boldsymbol{y}_w, \boldsymbol{y}_l)$, parameters $\boldsymbol{\theta}_t$ at step $t$, learning rate $\eta$, policy model $\pi_{\boldsymbol{\theta}_t}$
2: $z_{w,t}, z_{l,t} \leftarrow \log \pi_{\boldsymbol{\theta}_t}(\boldsymbol{y}_w \mid \boldsymbol{x}), \log \pi_{\boldsymbol{\theta}_t}(\boldsymbol{y}_l \mid \boldsymbol{x})$
3: *// — Start of Reward Calibration —*
4: Evaluate $d_{w,t}$ and $d_{l,t}$ for a given objective (Tab. 1)
5: $\|\boldsymbol{s}_{w,t}\|, \|\boldsymbol{s}_{l,t}\| \leftarrow \|\nabla_{\boldsymbol{\theta}_t} z_{w,t}\|, \|\nabla_{\boldsymbol{\theta}_t} z_{l,t}\|$
6: $r_t^\star, r_t \leftarrow \|\boldsymbol{s}_{l,t}\|/\|\boldsymbol{s}_{w,t}\|, d_{w,t}/d_{l,t}$
7: $z_{w,t}^{\mathrm{rc}} \leftarrow (r_t^\star/r_t)^{1/2} \cdot z_{w,t} + (1 - (r_t^\star/r_t)^{1/2}) \cdot \mathrm{sg}(z_{w,t})$
8: $z_{l,t}^{\mathrm{rc}} \leftarrow (r_t^\star/r_t)^{-1/2} \cdot z_{l,t} + (1 - (r_t^\star/r_t)^{-1/2}) \cdot \mathrm{sg}(z_{l,t})$
9: *// — End of Reward Calibration —*
10: Computing objective $\mathcal{L}(z_{w,t}^{\mathrm{rc}}, z_{l,t}^{\mathrm{rc}})$ for a given objective
11: $\boldsymbol{\theta}_{t+\eta} \leftarrow \boldsymbol{\theta}_t - \eta \nabla_{\boldsymbol{\theta}} \mathcal{L}(z_{w,t}^{\mathrm{rc}}, z_{l,t}^{\mathrm{rc}})$

---

## 4. Experimental Verifications

### 4.1. Experimental Settings

**Model and Benchmarks.** All models are trained for 1 epoch using AdamW with a constant learning rate (no warm-up) and a global batch size of 32. The learning rates are set per model scale and are consistent across baselines: Pythia-410M: $6e-7$, Pythia-2.8B: $1e-4$, Mistral-7B: $3e-7$, Qwen2.5-7B: $5e-7$. Experiments were run on two NVIDIA RTX 5090 D and six NVIDIA RTX A6000 GPUs.

We evaluate our RC method using the UltraFeedback Binarized dataset (Cui et al., 2024; Tunstall et al., 2024) and Anthropic Helpful and Harmless (Antropic-HH) (Bai et al., 2022). Our experiments employ Pythia-410M, Pythia-2.8B (Biderman et al., 2023), Mistral-7B-Base (Tunstall et al., 2024), and Qwen2.5-7B-Instruct (Yang et al., 2024) as the backbone architectures.

To assess robustness across different training regimes, we apply full-parameter fine-tuning on Pythia-410M and use QLoRA (Dettmers et al., 2023) for the other larger models. This setup ensures that our calibration method works across both full-weight updates and parameter-efficient fine-tuning.

We evaluate methods on the Open LLM Leaderboard (Gao et al., 2024), focusing on open-ended conversational alignment using AlpacaEval 2.0 (Dubois et al., 2024) with DeepSeek V3.2 (Liu et al., 2025), reporting the Length-Controlled Win Rate (LC Win Rate).

**Baselines and Implementation.** We compare our framework against two categories of baselines:

(1) Entangled objectives: margin-based methods that entangle the updates of chosen and rejected responses, such as DPO (Rafailov et al., 2023), IPO (Azar et al., 2024), and their variants CPO (Xu et al., 2024a), SimPO (Meng et al., 2024) and TI-DPO (Yang et al., 2026).

(2) Disentangled objectives: density ratio estimation methods that offer more independent control over updates, including DDRO (Higuchi & Suzuki, 2025) and the DIL family (BCE, LSIF, UKL) (Xiao et al., 2025).

## 4.2. Ablation Studies

We next examine two practical design choices in RC: approximating the score norms using output-side gradients, and stabilizing the ratio estimates with log-domain EMA.

**Efficient DB Estimation.** Computing the exact score norms $\|s_{w,t}\|, \|s_{l,t}\|$ requires a full backward pass through all trainable parameters. To make RC practical, we approximate them using output-side parameters: the language modeling head in full fine-tuning, and the final trainable adapter layers in parameter-efficient fine-tuning (e.g., LoRA (Hu et al., 2022) or QLoRA (Dettmers et al., 2023)). On Mistral-7B, the resulting DB closely matches the full-parameter version in both width and trend (Fig. 5), supporting this efficient approximation.

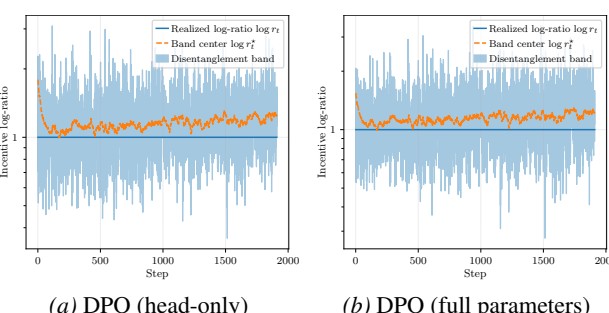

*(a)* DPO (head-only)      *(b)* DPO (full parameters)

*Figure 5.* Validation of the head-only gradient approximation on Mistral-7B. Head-only gradients produce a DB similar to that from full-parameter gradients in both width and trend.

**Sensitivity to EMA Momentum.** RC uses log-domain EMA to stabilize ratio estimates during stochastic training. We ablate the EMA momentum on Pythia-2.8B with DPO using $\beta \in \{0.5, 0.9, 0.95, 0.98, 0.999\}$. As shown in Tab. 3, performance is very similar across this range, indicating that RC is not sensitive to the EMA momentum and remains effective over a broad range of smoothing strengths.

*Table 3.* Ablation study on EMA momentum used in RC with DPO on Pythia-2.8B. RC remains effective across a wide range of EMA momenta. Subscripts denote standard errors.

| EMA | MMLU-Pro | BBH | MUSR | ARC | Average |
|---|---|---|---|---|---|
| w/o RC | $10.82_{.28}$ | $31.83_{.58}$ | $30.95_{1.61}$ | $34.04_{1.38}$ | 26.91 |
| 0.5 | $11.15_{.29}$ | $32.07_{2.85}$ | $33.86_{1.68}$ | $38.31_{1.42}$ | 28.85 |
| 0.9 | $10.97_{.28}$ | $32.54_{2.85}$ | $34.39_{1.68}$ | $37.63_{1.42}$ | 28.88 |
| 0.95 | $11.37_{.29}$ | $31.43_{2.81}$ | $33.47_{1.66}$ | $37.46_{1.41}$ | 28.43 |
| 0.98 | $11.15_{.29}$ | $32.12_{2.82}$ | $34.66_{1.68}$ | $37.88_{1.42}$ | 28.95 |
| 0.999 | $11.49_{.29}$ | $31.46_{2.82}$ | $34.52_{1.69}$ | $37.97_{1.42}$ | 28.86 |

**Training-Time Overhead of Reward Calibration.** RC is a lightweight backward-pass wrapper that introduces no auxiliary model or extra forward pass. Its wall-clock overhead is negligible under LoRA fine-tuning (Mistral-7B: $\sim$11h $\rightarrow$ $\sim$11.4h; Pythia-2.8B: $\sim$4.5h with little observed delay) and modest under full-parameter tuning on Pythia-410M ($\sim$50m $\rightarrow$ $\sim$59m). Detailed timings are reported in Tab. 8.

## 4.3. Experimental Results for Reward Calibration

**Verifying Pathway (iii) via Reward Calibration.** RC is designed to enforce Pathway (iii) by keeping the realized log-ratio $\log r_t$ inside the DB and near the center $\log r_t^\star$. On Pythia-2.8B (Fig. 4), entangled baselines align with the DB-based diagnosis: DPO/IPO frequently exhibit winner degradation together with loser suppression (Pathway (ii)), while CPO can instead drift toward joint increase (Pathway (i)). Importantly, objective form alone does not guarantee Pathway (iii): DDRO on Pythia-2.8B can also deviate from the DB-feasible regime and correspondingly fail to realize Pathway (iii) (see Fig. 10a in Sec. C.5). After applying RC, trajectories more often enter and remain in the Pathway (iii) regime across objectives and model scales from Pythia-410M to Mistral-7B (see Sec. C.5 for the full results).

**Downstream Performance Comparison On Benchmarks.** Tabs. 4 and 9 compares downstream performance with (w/) and without (w/o) RC. Overall, RC is more helpful when the uncalibrated run exhibits entangled or unstable likelihood dynamics, as rebalancing chosen vs. rejected updates facilitates training toward Pathway (iii).

On Pythia-2.8B, the clearest gains appear on ARC and BBH. For example, DPO improves ARC from 34.04 to 37.12 (+3.08), SimPO improves ARC from 32.59 to 37.03 (+4.44) and BBH from 30.49 to 31.34 (+0.85), and TI-DPO improves the average score from 27.13 to 28.44 (+1.31). In contrast, the effect is more task-dependent on MMLU-PRO and MUSR, where we observe both gains and regressions (e.g., MMLU-PRO improves for DPO/BCE/SimPO/TI-DPO but drops for CPO), consistent with RC being a dynamics-oriented intervention rather than a guaranteed monotonic booster.

On Mistral-7B, RC mainly acts as a stabilizer for weak runs. DPO improves from 24.26 to 36.44 on average (+12.18), with MMLU-Pro increasing from 11.66 to 30.31 and ARC from 22.70 to 60.92. DDRO shows a similar recovery, improving from 16.53 to 36.34 on average (+19.81). When the baseline is already strong, the changes are small, e.g., SimPO (+0.01), BCE (−0.05), LSIF (+0.15), and TI-DPO (+0.00) in average score. When the baseline is already strong (e.g., BCE/SimPO/LSIF), RC is non-disruptive, with changes typically near zero.

On Qwen2.5-7B, remains non-disruptive and brings modest improvements for most objectives. The average score improves for DPO (+0.48), SimPO (+0.26), BCE (+1.33), LSIF (+0.23), and TI-DPO (+0.11), while CPO is nearly unchanged (+0.01) and DDRO shows a small drop (−0.05). The gains are smaller than those on Mistral-7B, but the overall trend is still consistent with our main claim: promoting the Pathway (iii) regime is a useful and generally safe intervention, with clear gains in many entangled/unstable settings. See Sec. C.4 for detailed analysis.

*Table 4.* Evaluation results on benchmarks across models and objectives. Subscripts denote standard errors. For each objective, we report results without (w/o) RC and with (w/) RC. See Tab. 9 for the results on Pythia-2.8B.

| Method | RC | MMLU-Pro | BBH | MUSR | ARC | MATH | GSM8K | Average ($\Delta$) |
|---|---|---|---|---|---|---|---|---|
| | | | | **Mistral-7B** | | | | |
| DPO | w/o | $11.66_{.29}$ | $29.12_{2.65}$ | $41.53_{1.74}$ | $22.70_{1.22}$ | $2.93_{.46}$ | $37.60_{1.33}$ | 24.26 |
| | w/ | $30.31_{.42}$ | $44.83_{3.01}$ | $41.67_{1.74}$ | $60.92_{1.43}$ | $2.70_{.45}$ | $38.21_{1.34}$ | 36.44 (+12.18) |
| SimPO | w/o | $30.28_{.42}$ | $44.65_{3.01}$ | $41.80_{1.74}$ | $60.92_{1.43}$ | $2.80_{.46}$ | $37.68_{1.33}$ | 36.36 |
| | w/ | $30.19_{.42}$ | $45.03_{3.01}$ | $41.67_{1.74}$ | $61.18_{1.42}$ | $2.32_{.42}$ | $37.83_{1.34}$ | 36.37 (+0.01) |
| CPO | w/o | $30.12_{.42}$ | $44.73_{3.00}$ | $41.53_{1.75}$ | $60.24_{1.43}$ | $2.14_{.40}$ | $36.39_{1.33}$ | 35.86 |
| | w/ | $30.24_{.42}$ | $44.73_{3.01}$ | $41.53_{1.74}$ | $60.75_{1.43}$ | $2.74_{.45}$ | $37.53_{1.33}$ | 36.25 (+0.39) |
| BCE | w/o | $30.33_{.42}$ | $44.72_{3.01}$ | $41.67_{1.74}$ | $60.75_{1.43}$ | $3.03_{.47}$ | $37.83_{1.34}$ | 36.39 |
| | w/ | $30.26_{.42}$ | $44.84_{3.01}$ | $41.67_{1.74}$ | $60.84_{1.43}$ | $2.81_{.46}$ | $37.60_{1.33}$ | 36.34 (−0.05) |
| DDRO | w/o | $11.66_{.29}$ | $29.12_{2.65}$ | $35.71_{1.71}$ | $22.70_{1.22}$ | $0.00_{.00}$ | $0.00_{.00}$ | 16.53 |
| | w/ | $30.26_{.42}$ | $44.78_{3.01}$ | $41.40_{1.74}$ | $60.84_{1.43}$ | $2.63_{.44}$ | $38.13_{1.34}$ | 36.34 (+19.81) |
| LSIF | w/o | $30.30_{.42}$ | $44.80_{3.01}$ | $41.40_{1.74}$ | $60.75_{1.43}$ | $2.60_{.44}$ | $37.53_{1.33}$ | 36.23 |
| | w/ | $30.25_{.42}$ | $44.82_{3.00}$ | $41.40_{1.74}$ | $61.01_{1.43}$ | $2.81_{.46}$ | $37.98_{1.34}$ | 36.38 (+0.15) |
| TI-DPO | w/o | $30.35_{.42}$ | $45.02_{3.01}$ | $41.40_{1.74}$ | $60.58_{1.43}$ | $2.55_{1.13}$ | $37.30_{1.33}$ | 36.20 |
| | w/ | $30.30_{.42}$ | $44.95_{3.01}$ | $41.67_{1.74}$ | $60.49_{1.43}$ | $2.41_{1.02}$ | $37.38_{1.33}$ | 36.20 (+0.00) |
| | | | | **Qwen2.5-7B** | | | | |
| DPO | w/o | $44.32_{.45}$ | $55.74_{3.05}$ | $42.06_{1.76}$ | $67.49_{1.37}$ | $24.47_{3.02}$ | $34.72_{1.31}$ | 44.80 |
| | w/ | $44.79_{.45}$ | $55.76_{3.06}$ | $42.72_{1.78}$ | $67.32_{1.37}$ | $25.35_{3.07}$ | $35.71_{1.32}$ | 45.28 (+0.48) |
| SimPO | w/o | $44.82_{.45}$ | $55.50_{3.06}$ | $43.12_{1.78}$ | $67.49_{1.37}$ | $26.07_{3.10}$ | $36.09_{1.32}$ | 45.52 |
| | w/ | $44.86_{.45}$ | $55.64_{3.06}$ | $42.86_{1.78}$ | $67.49_{1.37}$ | $25.98_{3.05}$ | $37.83_{1.34}$ | 45.78 (+0.26) |
| CPO | w/o | $44.79_{.45}$ | $55.50_{3.06}$ | $43.52_{1.79}$ | $65.61_{1.39}$ | $26.20_{3.12}$ | $36.20_{1.38}$ | 45.30 |
| | w/ | $44.74_{.45}$ | $55.57_{3.06}$ | $43.25_{1.78}$ | $67.49_{1.37}$ | $25.42_{3.04}$ | $35.41_{1.32}$ | 45.31 (+0.01) |
| BCE | w/o | $41.33_{.45}$ | $53.90_{3.06}$ | $40.34_{1.74}$ | $67.41_{1.37}$ | $23.95_{2.97}$ | $35.41_{1.32}$ | 43.72 |
| | w/ | $44.77_{.45}$ | $55.61_{3.06}$ | $42.86_{1.78}$ | $67.32_{1.37}$ | $24.19_{2.99}$ | $35.56_{1.32}$ | 45.05 (+1.33) |
| DDRO | w/o | $44.80_{.45}$ | $55.71_{3.06}$ | $42.59_{1.78}$ | $67.24_{1.37}$ | $25.23_{3.04}$ | $35.94_{1.32}$ | 45.25 |
| | w/ | $44.80_{.45}$ | $55.71_{3.06}$ | $42.72_{1.78}$ | $67.24_{1.37}$ | $24.89_{3.04}$ | $35.86_{1.32}$ | 45.20 (−0.05) |
| LSIF | w/o | $44.78_{.45}$ | $55.68_{3.06}$ | $42.86_{1.78}$ | $67.32_{1.37}$ | $24.45_{3.02}$ | $35.63_{1.32}$ | 45.12 |
| | w/ | $44.80_{.45}$ | $55.65_{3.06}$ | $42.86_{1.78}$ | $67.24_{1.37}$ | $25.03_{3.04}$ | $36.54_{1.33}$ | 45.35 (+0.23) |
| TI-DPO | w/o | $44.76_{.45}$ | $55.55_{3.06}$ | $42.72_{1.78}$ | $67.15_{1.37}$ | $24.37_{3.00}$ | $35.86_{1.32}$ | 45.07 |
| | w/ | $44.77_{.45}$ | $55.66_{3.06}$ | $42.72_{1.78}$ | $67.24_{1.37}$ | $24.51_{3.04}$ | $36.16_{1.32}$ | 45.18 (+0.11) |

**Downstream Performance Comparison On Open-Ended Generation.** To assess open-ended generation, we further evaluate RC-calibrated Pythia-2.8B models against baselines on AlpacaEval 2.0 judged by DeepSeek V3.2 (Liu et al., 2025). RC improves the entangled DPO variant to a **70.56%** win rate (similar to 70.12% via GPT-4-Turbo[1]), while remaining non-regressive for the already-disentangled BCE variant (51.43%). This suggests that RC is particularly helpful for entangled objectives, while remaining non-regressive for the disentangled BCE variant in Pathway (iii) in this experiment.

*Table 5.* Evaluation results on Anthropic Helpful and Harmless (Anthropic-HH) benchmarks. Subscripts denote standard errors.

| Method | RC | MMLU-Pro | BBH | MUSR | ARC | MATH | GSM8K | Average |
|---|---|---|---|---|---|---|---|---|
| DPO | w/o | $30.14_{.42}$ | $44.82_{.61}$ | $41.40_{1.74}$ | $60.32_{1.43}$ | $2.42_{.43}$ | $37.15_{1.33}$ | 36.04 |
| | w/ | $30.17_{.42}$ | $44.63_{.61}$ | $41.67_{1.74}$ | $60.32_{1.43}$ | $2.17_{.42}$ | $36.62_{1.33}$ | 35.93 |
| BCE | w/o | $30.14_{.42}$ | $44.70_{.61}$ | $41.53_{1.74}$ | $60.41_{1.43}$ | $2.22_{.43}$ | $36.92_{1.33}$ | 35.99 |
| | w/ | $30.20_{.42}$ | $44.73_{.61}$ | $41.53_{1.74}$ | $60.49_{1.43}$ | $2.49_{.44}$ | $37.07_{1.33}$ | 36.09 |

[1] https://openai.com/api/

# 5. Conclusion and Future Works

**Conclusion.** We presented a unified framework for diagnosing and steering the reward dynamics of preference optimization. Through an incentive-score decomposition, we showed that diverse objectives share identical update directions and differ only in scalar incentive coefficients. This decomposition itself provides a common analytical lens for comparing objectives that have previously been studied in separate frameworks. This led to the disentanglement band (DB), a testable condition that characterizes when training enters the Pathway (iii) regime: suppress the loser while preserving the winner, possibly after an early stage. To enforce this condition, we introduced reward calibration (RC), a plug-and-play wrapper that rebalances chosen vs. rejected updates without redesigning the base objective. Empirically, RC often steers training toward Pathway (iii), with downstream improvements observed in several settings across diverse objectives and model scales.

**Limitations and Future Work.** Future work includes exploring other objective-agnostic ways to encourage Pathway (iii), extending the framework to iterative or online preference optimization, and generalizing the current pairwise setting to multi-response preference data.

## Acknowledgements

This work was supported in part by grants from National Natural Science Foundation of China (52539005), the China Scholarship Council (202306150167), the fundamental research program of Guangdong, China (2023A1515011281), Guangdong Basic and Applied Basic Research Foundation (24202107190000687), Foshan Science and Technology Research Project (2220001018608).

## Impact Statement

This paper presents work whose goal is to advance the field of preference optimization. While direct societal impacts are limited, future extensions to applied domains (e.g., via our open-source codebase) should incorporate domain-specific ethical reviews per deployment contexts.

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

# Appendix

# A. Notations and Preliminaries

### A.1. Notations

In this section, we summarize all the symbols used in this paper. See Tab. 6 for details.

*Table 6.* Notation Summary.

| Symbol | Definition | Notes |
|---|---|---|
| $(\boldsymbol{x}, \boldsymbol{y}_w, \boldsymbol{y}_l)$ | Preference triplet | $\boldsymbol{x}$: prompt $\boldsymbol{y}_w$: winner; $\boldsymbol{y}_l$: loser |
| $\pi_{\boldsymbol{\theta}}$ | Trainable policy model | Parameterized by $\boldsymbol{\theta}$ |
| $\pi_{\text{ref}}$ | Fixed reference policy model | Optional |
| $z_w(\boldsymbol{\theta})$ | $\log \pi_{\boldsymbol{\theta}}(\boldsymbol{y}_w \mid \boldsymbol{x})$ | Winner likelihood |
| $z_l(\boldsymbol{\theta})$ | $\log \pi_{\boldsymbol{\theta}}(\boldsymbol{y}_l \mid \boldsymbol{x})$ | Loser likelihood |
| $z_w^{\text{ref}}$ | $\log \pi_{\text{ref}}(\boldsymbol{y}_w \mid \boldsymbol{x})$ | Reference winner likelihoods |
| $z_l^{\text{ref}}$ | $\log \pi_{\text{ref}}(\boldsymbol{y}_l \mid \boldsymbol{x})$ | Reference loser likelihoods |
| $\tilde{z}_w(\boldsymbol{\theta})$ | $z_w(\boldsymbol{\theta}) - z_w^{\text{ref}}$ | Reference-normalized likelihoods |
| $\tilde{z}_l(\boldsymbol{\theta})$ | $z_l(\boldsymbol{\theta}) - z_l^{\text{ref}}$ | Reference-normalized likelihoods |
| $m(\boldsymbol{\theta})$ | $z_w(\boldsymbol{\theta}) - z_l(\boldsymbol{\theta})$ | Margin |
| $m_{\text{ref}}$ | $z_w^{\text{ref}} - z_l^{\text{ref}}$ | Reference margin |
| $\tilde{m}(\boldsymbol{\theta})$ | $m(\boldsymbol{\theta}) - m_{\text{ref}}$ | Effective margin |
| $\boldsymbol{s}_w(\boldsymbol{\theta})$ | $\nabla_{\boldsymbol{\theta}} \log \pi_{\boldsymbol{\theta}}(\boldsymbol{y}_w \mid \boldsymbol{x})$ | Score vectors |
| $\boldsymbol{s}_l(\boldsymbol{\theta})$ | $\nabla_{\boldsymbol{\theta}} \log \pi_{\boldsymbol{\theta}}(\boldsymbol{y}_l \mid \boldsymbol{x})$ | Score vectors |
| $\boldsymbol{\theta}_t$ | Continuous parameter trajectory | Trainable parameters at $t$ |
| $m_t$ | $m(\boldsymbol{\theta}_t)$ | Continuous-time margin at $t$ |

### A.2. Supplementary Related Works

**Density Ratio Estimation.** Density ratio estimation (DRE) is a classical statistical problem in machine learning (Sugiyama et al., 2012). Early discriminative/contrastive estimators include KLIEP (Sugiyama et al., 2008) and NCE (Gutmann & Hyvärinen, 2010; 2012), but they can be unstable under limited support overlap (Liu et al., 2017). More recent score-based DRE introduces intermediate distributions, ranging from discrete TRE (Rhodes et al., 2020) to continuous path formulations such as DRE-$\infty$ (Choi et al., 2021), with refinements via diffusion-bridge interpolants (Chen et al., 2025b), adaptive iterative paths based on the minimum variance path principle (Chen et al., 2026a), conditional probability path (Yu et al., 2025), and one-step score-based estimators (Chen et al., 2026b). The broader field of probabilistic inference (Paisley & Carin, 2009; Xu & Zeng, 2024) and Gaussian processes (Xu et al., 2024c;b), robust logical and semantic capabilities (Zhang et al., 2026; Luo et al., 2026; Bao et al., 2025) and generative modeling (Chen et al., 2025a; Xu et al., 2025) has also advanced sampling and inference techniques that underpin several of these estimators for capturing complex data structures (Li et al., 2023; 2025b; Lin et al., 2025). This literature provides context for density ratio-motivated preference objectives (e.g., DIL (Xiao et al., 2025), DDRO (Higuchi & Suzuki, 2025)). In contrast, we do not design new DRE estimators but analyze when such shaping yields disentangled likelihood dynamics and propose a plug-and-play intervention.

### A.3. Incentive Coefficients for Entangled and Disentangled Objectives

In this appendix we derive the incentive coefficients $(d_w, d_l)$ that appear in the unified update

$$-\nabla_{\boldsymbol{\theta}} \mathcal{L}(z_w(\boldsymbol{\theta}), z_l(\boldsymbol{\theta})) = -\frac{\partial \mathcal{L}}{\partial z_w} \nabla_{\boldsymbol{\theta}} z_w(\boldsymbol{\theta}) - \frac{\partial \mathcal{L}}{\partial z_l} \nabla_{\boldsymbol{\theta}} z_l(\boldsymbol{\theta}) \triangleq d_w(\boldsymbol{\theta}) \boldsymbol{s}_w(\boldsymbol{\theta}) - d_l(\boldsymbol{\theta}) \boldsymbol{s}_l(\boldsymbol{\theta}). \quad (13)$$

We define incentive coefficients as

$$d_w \triangleq -\frac{\partial \mathcal{L}}{\partial z_w}, \qquad d_l \triangleq \frac{\partial \mathcal{L}}{\partial z_l}. \quad (14)$$

For derivations, it is convenient to distinguish objectives by whether they are *entangled* (the loss ties $z_w$ and $z_l$ through a shared scalar, coupling $d_w$ and $d_l$) or *disentangled* (separate shaping of chosen vs. rejected terms, allowing more independent $d_w$ and $d_l$). This structural distinction on $(d_w, d_l)$ should not be conflated with density ratio-based vs. margin-based design (e.g., DIL (Xiao et al., 2025) vs. KTO (Ethayarajh et al., 2024; Yuan et al., 2025)). When a reference policy $\pi_{\text{ref}}$ is used, $z_w^{\text{ref}}, z_l^{\text{ref}}$ are treated as constants.

### A.3.1. ENTANGLED OBJECTIVES

We consider entangled objectives of the form (Yuan et al., 2025)

$$\mathcal{L}(z_w, z_l) = \ell\left(h_w(z_w) - h_l(z_l) - c\right) + \Lambda(z_w), \tag{15}$$

where $\ell(\cdot)$ is an outer shaping function, $h_w, h_l$ are transformations, $c$ is a constant offset (e.g., induced by a reference policy or length correction), and $\Lambda(z_w)$ is an optional chosen-only term (e.g., a likelihood regularizer in SlicHF (Zhao et al., 2023)). Substituting Eq. (15) into the definitions in Eq. (14) and applying the chain rule, we obtain

$$\begin{aligned} d_w &= -\ell'\left(h_w(z_w) - h_l(z_l) - c\right) h_w'(z_w) - \Lambda'(z_w), \\ d_l &= \ell'\left(h_w(z_w) - h_l(z_l) - c\right) h_l'(z_l), \end{aligned} \tag{16}$$

where $\ell'$ denotes the derivative of $\ell$ with respect to its scalar argument, and $h_w', h_l'$, and $\Lambda'$ denote the derivatives of $h_w, h_l$, and $\Lambda$ with respect to their respective inputs ($z_w$ or $z_l$). This notation is used throughout this subsection.

Below we instantiate Eq. (16) for common objectives.

**DPO.** DPO (Rafailov et al., 2023) uses $\mathcal{L}_{\text{DPO}}(z_w, z_l) = -\log\sigma\left(\beta\left((z_w - z_l) - (z_w^{\text{ref}} - z_l^{\text{ref}})\right)\right)$. This corresponds to $h_w(z_w) = \beta z_w$, $h_l(z_l) = \beta z_l$, $c = \beta(z_w^{\text{ref}} - z_l^{\text{ref}})$, and $\Lambda(z_w) \equiv 0$. Since $\frac{\mathrm{d}}{\mathrm{d}u}\left[-\log\sigma(u)\right] = -\sigma(-u)$, we obtain

$$d_w = d_l = \beta\sigma\left(-\beta\tilde{m}\right), \tag{17}$$

where $\tilde{m} = (z_w - z_l) - (z_w^{\text{ref}} - z_l^{\text{ref}})$.

**IPO.** IPO (Azar et al., 2024, identity preference optimization) uses $\mathcal{L}_{\text{IPO}}(z_w, z_l) = (\tilde{m} - \lambda)^2$. Here $h_w(z_w) = z_w$, $h_l(z_l) = z_l$, $c = z_w^{\text{ref}} - z_l^{\text{ref}}$, $\Lambda(z_w) \equiv 0$, and $\ell(u) = (u - \lambda)^2$, hence $\ell'(u) = 2(u - \lambda)$. Therefore, we have

$$d_w = d_l = 2(\lambda - \tilde{m}). \tag{18}$$

**R-DPO.** R-DPO (Park et al., 2024, regularized-DPO) modifies the DPO argument by a length-difference correction: $\mathcal{L}_{\text{R-DPO}}(z_w, z_l) = -\log\sigma(\beta\tilde{m} + \alpha\Delta|\boldsymbol{y}|)$, where $\Delta|\boldsymbol{y}| = |\boldsymbol{y}_w| - |\boldsymbol{y}_l|$. Here, $h_w(z_w) = \beta z_w$, $h_l(z_l) = \beta z_l$, $\Lambda(z_w) \equiv 0$, and a modified offset $c = \beta(z_w^{\text{ref}} - z_l^{\text{ref}}) - \alpha\Delta|\boldsymbol{y}|$. The resulting incentive coefficients are

$$d_w = d_l = \beta\sigma\left(-\beta\tilde{m} - \alpha\Delta|\boldsymbol{y}|\right). \tag{19}$$

**SimPO.** SimPO (Meng et al., 2024, simple preference optimization) uses a length-normalized margin $m_{\text{norm}} = \frac{\beta}{|\boldsymbol{y}_w|}z_w - \frac{\beta}{|\boldsymbol{y}_l|}z_l$ and loss $\mathcal{L}_{\text{SimPO}}(z_w, z_l) = -\log\sigma(m_{\text{norm}} - \gamma)$. With $h_w(z_w) = \frac{\beta}{|\boldsymbol{y}_w|}z_w$, $h_l(z_l) = \frac{\beta}{|\boldsymbol{y}_l|}z_l$, $c = \gamma$, and $\Lambda(z_w) \equiv 0$, we get

$$d_w = \frac{\beta}{|\boldsymbol{y}_w|}\sigma(\gamma - m_{\text{norm}}), \qquad d_l = \frac{\beta}{|\boldsymbol{y}_l|}\sigma(\gamma - m_{\text{norm}}). \tag{20}$$

**CPO.** CPO (Xu et al., 2024a, contrastive preference optimization) augments the standard margin loss with a chosen-only likelihood term: $\mathcal{L}_{\text{CPO}}(z_w, z_l) = -\log\sigma(\beta m) - \lambda z_w$, where $m = z_w - z_l$. This corresponds to $h_w(z_w) = \beta z_w$, $h_l(z_l) = \beta z_l$, and $c = 0$. Crucially, it sets $\Lambda(z_w) = -\lambda z_w$, so that minimizing $\mathcal{L}_{\text{CPO}}$ encourages larger $z_w$. Therefore

$$d_w = \beta\sigma(-\beta m) + \lambda, \qquad d_l = \beta\sigma(-\beta m). \tag{21}$$

**RRHF / SlicHF.** Hinge-style objectives such as RRHF (Yuan et al., 2023) and SlicHF (Zhao et al., 2023) use

$$\mathcal{L}_{\text{hinge}}(z_w, z_l) = \max(0, \gamma - m) - \lambda z_w, \qquad m = z_w - z_l, \tag{22}$$

where $\gamma = 0$ recovers RRHF. Here $h_w(z_w) = z_w$, $h_l(z_l) = z_l$, $c = 0$, and $\Lambda(z_w) = -\lambda z_w$. Using a subgradient for the hinge term, $\frac{\partial}{\partial m} \max(0, \gamma - m) = -\mathbb{I}(m < \gamma)$, we obtain

$$d_l = \mathbb{I}(m < \gamma), \qquad d_w = \mathbb{I}(m < \gamma) + \lambda. \tag{23}$$

**TDPO.** For TDPO (Zeng et al., 2024, token-level DPO), we keep the winner-side statistic unchanged and absorb the rejected-side token-level correction into the loser-side statistic: $z_l \triangleq \log \pi_{\boldsymbol{\theta}}(\boldsymbol{y}_l \mid \boldsymbol{x}) + \alpha D_{\text{SeqKL}}(\boldsymbol{x}, \boldsymbol{y}_l; \pi_{\text{ref}} \| \pi_{\boldsymbol{\theta}})$, where

$$D_{\text{SeqKL}}(\boldsymbol{x}, \boldsymbol{y}; \pi_{\text{ref}} \| \pi_{\boldsymbol{\theta}}) = \sum_{t=1}^{|\boldsymbol{y}|} D_{\text{KL}}\left(\pi_{\text{ref}}(\cdot \mid [\boldsymbol{x}, \boldsymbol{y}_{<t}]) \| \pi_{\boldsymbol{\theta}}(\cdot \mid [\boldsymbol{x}, \boldsymbol{y}_{<t}])\right). \tag{24}$$

Define $D_w \triangleq D_{\text{SeqKL}}(\boldsymbol{x}, \boldsymbol{y}_w; \pi_{\text{ref}} \| \pi_{\boldsymbol{\theta}})$, then TDPO can be written as $\mathcal{L}_{\text{TDPO}}(z_w, z_l) = -\log \sigma(\beta[\tilde{m} + \alpha \, \text{sg}(D_w)])$. Therefore,

$$d_w = -\frac{\partial \mathcal{L}_{\text{TDPO}}}{\partial z_w} = \beta \sigma(-\beta[\tilde{m} + \alpha \, \text{sg}(D_w)]), \qquad d_l = \frac{\partial \mathcal{L}_{\text{TDPO}}}{\partial z_l} = \beta \sigma(-\beta[\tilde{m} + \alpha \, \text{sg}(D_w)]). \tag{25}$$

The corresponding scores are $\boldsymbol{s}_w = \nabla_{\boldsymbol{\theta}} \log \pi_{\boldsymbol{\theta}}(\boldsymbol{y}_w \mid \boldsymbol{x})$ and $\boldsymbol{s}_l = \nabla_{\boldsymbol{\theta}} \log \pi_{\boldsymbol{\theta}}(\boldsymbol{y}_l \mid \boldsymbol{x}) + \alpha \nabla_{\boldsymbol{\theta}} D_{\text{SeqKL}}(\boldsymbol{x}, \boldsymbol{y}_l; \pi_{\text{ref}} \| \pi_{\boldsymbol{\theta}})$. Hence TDPO still admits the same two-branch decomposition $-\nabla_{\boldsymbol{\theta}} \mathcal{L}_{\text{TDPO}} = d_w \boldsymbol{s}_w - d_l \boldsymbol{s}_l$. Therefore, TDPO remains entangled: it preserves the symmetric incentive coefficients $d_w = d_l$, while modifying the loser-side score direction through an additional token-level sequential KL term.

**TI-DPO.** For the weighted token-level preference term in TI-DPO (Yang et al., 2026, token-importance DPO), define the winner/loser branch statistics as weighted sums of token-level log-probabilities:

$$z_w \triangleq \sum_{t=1}^{|\boldsymbol{y}_w|} \omega_{w,t} \log \pi_{\boldsymbol{\theta}}(y_{w,t} \mid [\boldsymbol{x}, \boldsymbol{y}_{w,<t}]), \qquad z_l \triangleq \sum_{t=1}^{|\boldsymbol{y}_l|} \omega_{l,t} \log \pi_{\boldsymbol{\theta}}(y_{l,t} \mid [\boldsymbol{x}, \boldsymbol{y}_{l,<t}]), \tag{26}$$

where $\omega_{w,t}$ and $\omega_{l,t}$ are the token-importance weights. Define $z_w^{\text{ref}}$ and $z_l^{\text{ref}}$ analogously by replacing $\pi_{\boldsymbol{\theta}}$ with $\pi_{\text{ref}}$. Then the weighted DPO term in TI-DPO takes the standard entangled form $\mathcal{L}_{\text{TI-DPO}} = -\log \sigma(\beta \tilde{m})$. Its incentive coefficients are therefore

$$d_w = \beta \sigma(-\beta \tilde{m}), \qquad d_l = \beta \sigma(-\beta \tilde{m}). \tag{27}$$

The corresponding score directions become token-weighted sums:

$$\boldsymbol{s}_w = \sum_{t=1}^{|\boldsymbol{y}_w|} \omega_{w,t} \nabla_{\boldsymbol{\theta}} \log \pi_{\boldsymbol{\theta}}(y_{w,t} \mid [\boldsymbol{x}, \boldsymbol{y}_{w,<t}]), \qquad \boldsymbol{s}_l = \sum_{t=1}^{|\boldsymbol{y}_l|} \omega_{l,t} \nabla_{\boldsymbol{\theta}} \log \pi_{\boldsymbol{\theta}}(y_{l,t} \mid [\boldsymbol{x}, \boldsymbol{y}_{l,<t}]). \tag{28}$$

Therefore, the token-level weighting changes the branch statistics and the resulting score directions, but not the form of the incentive coefficients.

### A.3.2. DISENTANGLED OBJECTIVES

Disentangled objectives decompose into independent chosen and rejected terms, typically through reference-normalized log density ratios $\tilde{z}_w \triangleq z_w - z_w^{\text{ref}}$ and $\tilde{z}_l \triangleq z_l - z_l^{\text{ref}}$. In entangled objectives, there is only one outer shaping function $\ell$, while a general disentangled form is

$$\mathcal{L}(z_w, z_l) = \ell_w(\tilde{z}_w) + \ell_l(\tilde{z}_l), \tag{29}$$

for which

$$d_w = -\ell_w'(\tilde{z}_w) = -\frac{\partial}{\partial \tilde{z}_w} \ell_w(\tilde{z}_w), \qquad d_l = \ell_l'(\tilde{z}_l) = \frac{\partial}{\partial \tilde{z}_l} \ell_l(\tilde{z}_l). \tag{30}$$

Here $\ell_w$ and $\ell_l$ are outer shaping functions corresponding to $z_w$ and $z_l$, respectively.

We now derive the incentive coefficients $(d_w, d_l)$ for disentangled cases.

**KTO.** KTO (Ethayarajh et al., 2024, Kahneman-Tversky optimization) applies pointwise shaping in log-probability space:

$$\mathcal{L}_{\text{KTO}}(z_w, z_l) = \lambda_w \sigma(z_w^{\text{ref}} - z_w) + \lambda_l \sigma(z_l - z_l^{\text{ref}}). \tag{31}$$

Let $\sigma'(u) \triangleq \sigma(u)(1 - \sigma(u))$. Then

$$d_w = \lambda_w \sigma'\left(z_w^{\text{ref}} - z_w\right) = \lambda_w \sigma'(-\tilde{z}_w), \qquad d_l = \lambda_l \sigma'\left(z_l - z_l^{\text{ref}}\right) = \lambda_l \sigma'(\tilde{z}_l), \tag{32}$$

so KTO satisfies $d_w \neq d_l$ in general.

**DDRO.** DDRO ((Higuchi & Suzuki, 2025), direct density ratio optimization) operates on an unpaired dataset introduced by KTO. The resulting objective of DDRO is defined as:

$$\mathcal{L}_{\text{DDRO}}(z_w, z_l) = \log 2 - \tilde{z}_w + \log 2 - \log\left(2 - e^{\tilde{z}_l}\right), \tag{33}$$

where $\ell_w(\tilde{z}_w) = \log 2 - \tilde{z}_w$ and $\ell_l(\tilde{z}_l) = \log 2 - \log\left(2 - e^{\tilde{z}_l}\right)$. Then, we have

$$d_w = 1, \qquad d_l = \frac{e^{\tilde{z}_l}}{2 - e^{\tilde{z}_l}}. \tag{34}$$

**DIL-BCE.** The BCE (Hastie et al., 2001) variant in DIL (Xiao et al., 2025) is

$$\mathcal{L}_{\text{DIL-BCE}}(z_w, z_l) = \log\left(1 + e^{-\tilde{z}_w}\right) + \log\left(1 + e^{\tilde{z}_l}\right). \tag{35}$$

Using $\frac{\mathrm{d}}{\mathrm{d}u} \log(1 + e^{-u}) = -\sigma(-u)$ and $\frac{\mathrm{d}}{\mathrm{d}u} \log(1 + e^u) = \sigma(u)$, we obtain

$$d_w = \sigma(-\tilde{z}_w), \qquad d_l = \sigma(\tilde{z}_l). \tag{36}$$

**DIL-UKL.** The UKL (Nguyen et al., 2010) variant in DIL (Xiao et al., 2025) uses $\mathcal{L}_{\text{DIL-UKL}}(z_w, z_l) = e^{\tilde{z}_l} - \tilde{z}_w$, yielding

$$d_w = 1, \qquad d_l = e^{\tilde{z}_l}. \tag{37}$$

**DIL-LSIF.** The LSIF (Kanamori et al., 2009) variant in DIL (Xiao et al., 2025) uses $\mathcal{L}_{\text{DIL-LSIF}}(z_w, z_l) = \frac{1}{2} e^{2\tilde{z}_l} - e^{\tilde{z}_w}$, yielding

$$d_w = e^{\tilde{z}_w}, \qquad d_l = e^{2\tilde{z}_l}. \tag{38}$$

## B. Proofs

### B.1. Proof of Theorem 3.1

**Theorem 3.1.** *Consider a parameter trajectory $\boldsymbol{\theta}_t$ driven by the gradient flow of Eq. (5). Along $\boldsymbol{\theta}_t$, define $z_{w|l,t} \triangleq z_{w|l}(\boldsymbol{\theta}_t)$, $\boldsymbol{s}_{w|l,t} \triangleq \nabla_{\boldsymbol{\theta}} z_{w|l}(\boldsymbol{\theta}_t)$, and $d_{w|l,t} \triangleq d_{w|l}(\boldsymbol{\theta}_t)$, with $w|l$ being $w$ or $l$. Then, the continuous-time dynamics of the chosen and rejected likelihoods are governed by:*

$$\begin{aligned} \frac{\partial z_{w,t}}{\partial t} &= \dot{z}_{w,t} = d_{w,t}\|\boldsymbol{s}_{w,t}\|^2 - d_{l,t}\langle \boldsymbol{s}_{w,t}, \boldsymbol{s}_{l,t}\rangle, \\ \frac{\partial z_{l,t}}{\partial t} &= \dot{z}_{l,t} = d_{w,t}\langle \boldsymbol{s}_{w,t}, \boldsymbol{s}_{l,t}\rangle - d_{l,t}\|\boldsymbol{s}_{l,t}\|^2. \end{aligned} \tag{6}$$

*Proof of Theorem 3.1.* Fix a time $t$ and consider the infinitesimal evolution induced by the local update direction at $\boldsymbol{\theta}_t$.

By the assumed differentiability of $z_w(\boldsymbol{\theta})$ and $z_l(\boldsymbol{\theta})$, first-order Taylor expansions around $\boldsymbol{\theta}_t$ yield

$$z_{w,t+\eta} = z_{w,t} + \langle \nabla_{\boldsymbol{\theta}} z_w(\boldsymbol{\theta}_t), \boldsymbol{\theta}_{t+\eta} - \boldsymbol{\theta}_t\rangle + \mathcal{O}\left(\|\boldsymbol{\theta}_{t+\eta} - \boldsymbol{\theta}_t\|^2\right), \tag{39}$$

$$z_{l,t+\eta} = z_{l,t} + \langle \nabla_{\boldsymbol{\theta}} z_l(\boldsymbol{\theta}_t), \boldsymbol{\theta}_{t+\eta} - \boldsymbol{\theta}_t\rangle + \mathcal{O}\left(\|\boldsymbol{\theta}_{t+\eta} - \boldsymbol{\theta}_t\|^2\right). \tag{40}$$

Introduce the score vectors at time $t$, $s_{w|l,t} \triangleq \nabla_{\theta} z_{w|l}(\theta_t)$. With this notation, the linear terms in Eqs. (39) and (40) become $\langle s_{w,t}, \theta_{t+\eta} - \theta_t \rangle$ and $\langle s_{l,t}, \theta_{t+\eta} - \theta_t \rangle$ respectively. Consider a gradient descent discretization with step size $\eta > 0$,

$$\theta_{t+\eta} - \theta_t = -\eta \nabla_{\theta} \mathcal{L}(\theta_t) = \eta \left( d_{w,t} s_{w,t} - d_{l,t} s_{l,t} \right), \tag{41}$$

where the second equation can be derived by the unified gradient form evaluated at $\theta_t$, $-\nabla_{\theta} \mathcal{L}(\theta_t) = d_{w,t} s_{w,t} - d_{l,t} s_{l,t}$, as defined in Eq. (5). In particular, This implies $\|\theta_{t+\eta} - \theta_t\| = \mathcal{O}(\eta)$ as $\eta \to 0$, so the Taylor remainders in Eqs. (39) and (40) satisfy $\mathcal{O}(\|\theta_{t+\eta} - \theta_t\|^2) = \mathcal{O}(\eta^2)$.

Substituting Eq. (41) into the linear terms in Eqs. (39) and (40) gives

$$\begin{aligned} z_{w,t+\eta} - z_{w,t} &= \langle s_{w,t}, \ \eta \left( d_{w,t} s_{w,t} - d_{l,t} s_{l,t} \right) \rangle + \mathcal{O}(\eta^2) \\ &= \eta \left( d_{w,t} \langle s_{w,t}, s_{w,t} \rangle - d_{l,t} \langle s_{w,t}, s_{l,t} \rangle \right) + \mathcal{O}(\eta^2) \\ &= \eta \left( d_{w,t} \|s_{w,t}\|^2 - d_{l,t} \langle s_{w,t}, s_{l,t} \rangle \right) + \mathcal{O}(\eta^2), \end{aligned} \tag{42}$$

$$\begin{aligned} z_{l,t+\eta} - z_{l,t} &= \langle s_{l,t}, \ \eta \left( d_{w,t} s_{w,t} - d_{l,t} s_{l,t} \right) \rangle + \mathcal{O}(\eta^2) \\ &= \eta \left( d_{w,t} \langle s_{l,t}, s_{w,t} \rangle - d_{l,t} \langle s_{l,t}, s_{l,t} \rangle \right) + \mathcal{O}(\eta^2) \\ &= \eta \left( d_{w,t} \langle s_{w,t}, s_{l,t} \rangle - d_{l,t} \|s_{l,t}\|^2 \right) + \mathcal{O}(\eta^2), \end{aligned} \tag{43}$$

where we used $\langle s_{l,t}, s_{w,t} \rangle = \langle s_{w,t}, s_{l,t} \rangle$.

Divide Eqs. (42) and (43) by $\eta$ and take $\eta \to 0$ to derive the instantaneous dynamics:

$$\dot{z}_{w,t} = \lim_{\eta \to 0} \frac{z_{w,t+\eta} - z_{w,t}}{\eta} = d_{w,t} \|s_{w,t}\|^2 - d_{l,t} \langle s_{w,t}, s_{l,t} \rangle + \lim_{\eta \to 0} \mathcal{O}(\eta) = d_{w,t} \|s_{w,t}\|^2 - d_{l,t} \langle s_{w,t}, s_{l,t} \rangle, \tag{44}$$

$$\dot{z}_{l,t} = \lim_{\eta \to 0} \frac{z_{l,t+\eta} - z_{l,t}}{\eta} = d_{w,t} \langle s_{w,t}, s_{l,t} \rangle - d_{l,t} \|s_{l,t}\|^2 + \lim_{\eta \to 0} \mathcal{O}(\eta) = d_{w,t} \langle s_{w,t}, s_{l,t} \rangle - d_{l,t} \|s_{l,t}\|^2. \tag{45}$$

which are exactly Eq. (6). $\qquad\square$

### B.2. Proof of Corollary 3.2

**Corollary 3.2.** *Define the time-indexed margin $m_t \triangleq z_{w,t} - z_{l,t}$. Then, the continuous-time dynamic of $m_t$ satisfy*

$$\dot{m}_t = d_{w,t} \|s_{w,t}\|^2 + d_{l,t} \|s_{l,t}\|^2 - (d_{w,t} + d_{l,t}) \langle s_{w,t}, s_{l,t} \rangle.$$

*Proof of Corollary 3.2.* By definition, $m_t = z_{w,t} - z_{l,t}$, hence $\dot{m}_t = \dot{z}_{w,t} - \dot{z}_{l,t}$. Substituting the dynamics in Eq. (6), we obtain

$$\begin{aligned} \dot{m}_t &= \left( d_{w,t} \|s_{w,t}\|^2 - d_{l,t} \langle s_{w,t}, s_{l,t} \rangle \right) - \left( d_{w,t} \langle s_{w,t}, s_{l,t} \rangle - d_{l,t} \|s_{l,t}\|^2 \right) \\ &= d_{w,t} \|s_{w,t}\|^2 + d_{l,t} \|s_{l,t}\|^2 - (d_{w,t} + d_{l,t}) \langle s_{w,t}, s_{l,t} \rangle. \end{aligned}$$

This finishes the proof. $\qquad\square$

### B.3. Proof of Proposition 3.3

**Proposition 3.3.** *For given $\rho_t$ and $\|s_{l,t}\|/\|s_{w,t}\|$, the maximizer of $\Delta_t(\log r_t)$ over all $\log r_t$ inside DB satisfies*

$$\log r_t^{\star} = \arg \max_{\log r_t \in DB} \Delta_t(\log r_t) = \log \frac{\|s_{l,t}\|}{\|s_{w,t}\|}. \tag{11}$$

*Proof of Proposition 3.3.* Recall the DB written in log-ratio form:

$$\log \frac{\|s_{l,t}\|}{\|s_{w,t}\|} + \log \rho_t \le \log r_t \le \log \frac{\|s_{l,t}\|}{\|s_{w,t}\|} - \log \rho_t. \tag{46}$$

Define the two DB thresholds in ratio space as

$$a_t \triangleq \rho_t \frac{\|s_{l,t}\|}{\|s_{w,t}\|}, \qquad b_t \triangleq \frac{1}{\rho_t} \frac{\|s_{l,t}\|}{\|s_{w,t}\|}, \tag{47}$$

so that $\log r_t \in \mathrm{DB}$ is equivalent to $r_t \in [a_t, b_t]$. Let $c_t \triangleq \sqrt{a_t b_t}$.

By definition,

$$\Delta_t(\log r_t) = \min \left\{ \log \frac{r_t}{a_t}, \log \frac{b_t}{r_t} \right\}. \tag{48}$$

**Inverval** $r_t \in [a_t, c_t]$. We first consider $r_t \in [a_t, c_t]$. Since $r_t \leq c_t$ and all quantities are positive,

$$r_t \leq \sqrt{a_t b_t} \implies r_t^2 \leq a_t b_t \implies \frac{r_t}{a_t} \leq \frac{b_t}{r_t} \implies \log \frac{r_t}{a_t} \leq \log \frac{b_t}{r_t}. \tag{49}$$

The last implication uses that $\log(\cdot)$ is strictly increasing on $(0, \infty)$. Therefore, on $[a_t, c_t]$ the minimum in Eq. (48) is attained by the first term, namely $\Delta_t(\log r_t) = \log \frac{r_t}{a_t}$ for all $r_t \in [a_t, c_t]$.

Moreover, for any $a_t \leq r_t < r'_t \leq c_t$,

$$\Delta_t(\log r'_t) - \Delta_t(\log r_t) = \log \frac{r'_t}{a_t} - \log \frac{r_t}{a_t} = \log \frac{r'_t}{r_t} > 0, \tag{50}$$

so $\Delta_t(\log r_t)$ is strictly increasing on $[a_t, c_t]$.

**Inverval** $r_t \in [c_t, b_t]$. Next consider $r_t \in [c_t, b_t]$. Since $r_t \geq c_t$,

$$r_t \geq \sqrt{a_t b_t} \implies r_t^2 \geq a_t b_t \implies \frac{r_t}{a_t} \geq \frac{b_t}{r_t} \implies \log \frac{r_t}{a_t} \geq \log \frac{b_t}{r_t}. \tag{51}$$

Hence, on $[c_t, b_t]$ the minimum in Eq. (48) is attained by the second term: $\Delta_t(\log r_t) = \log \frac{b_t}{r_t}$ for all $r_t \in [c_t, b_t]$.

For any $c_t \leq r_t < r'_t \leq b_t$,

$$\Delta_t(\log r'_t) - \Delta_t(\log r_t) = \log \frac{b_t}{r'_t} - \log \frac{b_t}{r_t} = \log \frac{r_t}{r'_t} < 0, \tag{52}$$

so $\Delta_t(\log r_t)$ is strictly decreasing on $[c_t, b_t]$.

Together, since $\Delta_t(\log r_t)$ is strictly increasing on $[a_t, c_t]$ and strictly decreasing on $[c_t, b_t]$, it has a unique maximizer on $[a_t, b_t]$ at $r_t = c_t$. Finally,

$$\log r_t^\star = \log c_t = \frac{1}{2} \big( \log a_t + \log b_t \big) = \log \frac{\|s_{l,t}\|}{\|s_{w,t}\|}. \tag{53}$$

This completes the proof.

$\square$

### B.4. Proof of Proposition 3.4

**Proposition 3.4.** *The log-ratio induced by the calibrated likelihoods in Eq.* (12) *is exactly the log-ratio* $\log r_t^\star$ *defined in Proposition* 3.3.

*Proof of Proposition 3.4.* Let $\mathrm{sg}(\cdot)$ denote the stop-gradient operator: for any scalar $x$, $\mathrm{sg}(x) = x$ in the forward pass while $\frac{\partial}{\partial x} \mathrm{sg}(x) = 0$ in the backward pass.

Fix time $t$. Let $r_t \triangleq d_{w,t}/d_{l,t}$ be the incentive ratio induced by the base objective $\mathcal{L}(z_w, z_l)$ at time $t$, and let $r_t^\star$ be the target ratio in Proposition 3.3. Define

$$\alpha_t \triangleq \left( \frac{r_t^\star}{r_t} \right)^{1/2} = \exp \left( \frac{1}{2} (\log r_t^\star - \log r_t) \right). \tag{54}$$

We construct calibrated likelihoods by

$$z_{w,t}^{\mathrm{rc}} \triangleq \alpha_t z_{w,t} + (1 - \alpha_t)\,\mathrm{sg}(z_{w,t}), \qquad z_{l,t}^{\mathrm{rc}} \triangleq \alpha_t^{-1} z_{l,t} + (1 - \alpha_t^{-1})\,\mathrm{sg}(z_{l,t}). \tag{55}$$

By construction, in the forward pass we have $z_{w,t}^{\mathrm{rc}} = z_{w,t}$ and $z_{l,t}^{\mathrm{rc}} = z_{l,t}$, but in the backward pass the Jacobians satisfy

$$\frac{\partial z_{w,t}^{\mathrm{rc}}}{\partial z_{w,t}} = \alpha_t, \qquad \frac{\partial z_{l,t}^{\mathrm{rc}}}{\partial z_{l,t}} = \alpha_t^{-1}, \qquad \frac{\partial z_{w,t}^{\mathrm{rc}}}{\partial z_{l,t}} = \frac{\partial z_{l,t}^{\mathrm{rc}}}{\partial z_{w,t}} = 0, \tag{56}$$

since $\frac{\partial}{\partial z}\,\mathrm{sg}(z) = 0$.

Now consider the composed objective $\mathcal{L}(z_{w,t}^{\mathrm{rc}}, z_{l,t}^{\mathrm{rc}})$, viewed as a function of $(z_{w,t}, z_{l,t})$ through the transformations above, and define the calibrated incentive coefficients by the same rule:

$$d_{w,t}^{\mathrm{rc}} \triangleq -\frac{\partial \mathcal{L}(z_{w,t}^{\mathrm{rc}}, z_{l,t}^{\mathrm{rc}})}{\partial z_{w,t}}, \qquad d_{l,t}^{\mathrm{rc}} \triangleq \frac{\partial \mathcal{L}(z_{w,t}^{\mathrm{rc}}, z_{l,t}^{\mathrm{rc}})}{\partial z_{l,t}}. \tag{57}$$

Applying the chain rule and using Eq. (56), we obtain

$$\frac{\partial \mathcal{L}(z_{w,t}^{\mathrm{rc}}, z_{l,t}^{\mathrm{rc}})}{\partial z_{w,t}} = \frac{\partial \mathcal{L}(z_{w,t}^{\mathrm{rc}}, z_{l,t}^{\mathrm{rc}})}{\partial z_{w,t}^{\mathrm{rc}}} \cdot \frac{\partial z_{w,t}^{\mathrm{rc}}}{\partial z_{w,t}} + \frac{\partial \mathcal{L}(z_{w,t}^{\mathrm{rc}}, z_{l,t}^{\mathrm{rc}})}{\partial z_{l,t}^{\mathrm{rc}}} \cdot \frac{\partial z_{l,t}^{\mathrm{rc}}}{\partial z_{w,t}} = \alpha_t \frac{\partial \mathcal{L}(z_{w,t}^{\mathrm{rc}}, z_{l,t}^{\mathrm{rc}})}{\partial z_{w,t}^{\mathrm{rc}}},$$

$$\frac{\partial \mathcal{L}(z_{w,t}^{\mathrm{rc}}, z_{l,t}^{\mathrm{rc}})}{\partial z_{l,t}} = \frac{\partial \mathcal{L}(z_{w,t}^{\mathrm{rc}}, z_{l,t}^{\mathrm{rc}})}{\partial z_{w,t}^{\mathrm{rc}}} \cdot \frac{\partial z_{w,t}^{\mathrm{rc}}}{\partial z_{l,t}} + \frac{\partial \mathcal{L}(z_{w,t}^{\mathrm{rc}}, z_{l,t}^{\mathrm{rc}})}{\partial z_{l,t}^{\mathrm{rc}}} \cdot \frac{\partial z_{l,t}^{\mathrm{rc}}}{\partial z_{l,t}} = \alpha_t^{-1} \frac{\partial \mathcal{L}(z_{w,t}^{\mathrm{rc}}, z_{l,t}^{\mathrm{rc}})}{\partial z_{l,t}^{\mathrm{rc}}}. \tag{58}$$

The following point is crucial: by Eq. (55), the numerical forward values satisfy $(z_{w,t}^{\mathrm{rc}}, z_{l,t}^{\mathrm{rc}}) = (z_{w,t}, z_{l,t})$. Therefore, the partial derivatives of $\mathcal{L}$ with respect to its first argument, evaluated at these two notations, coincide:

$$\frac{\partial \mathcal{L}(z_{w,t}^{\mathrm{rc}}, z_{l,t}^{\mathrm{rc}})}{\partial z_{w,t}^{\mathrm{rc}}}\Bigg|_{(z_{w,t}^{\mathrm{rc}}, z_{l,t}^{\mathrm{rc}})=(z_{w,t}, z_{l,t})} = \frac{\partial \mathcal{L}(z_{w,t}, z_{l,t})}{\partial z_{w,t}}\Bigg|_{(z_{w,t}, z_{l,t})},$$

$$\frac{\partial \mathcal{L}(z_{w,t}^{\mathrm{rc}}, z_{l,t}^{\mathrm{rc}})}{\partial z_{l,t}^{\mathrm{rc}}}\Bigg|_{(z_{w,t}^{\mathrm{rc}}, z_{l,t}^{\mathrm{rc}})=(z_{w,t}, z_{l,t})} = \frac{\partial \mathcal{L}(z_{w,t}, z_{l,t})}{\partial z_{l,t}}\Bigg|_{(z_{w,t}, z_{l,t})}. \tag{59}$$

Substituting Eqs. (58) and (59) into Eq. (57) yields

$$d_{w,t}^{\mathrm{rc}} = -\frac{\partial \mathcal{L}(z_{w,t}^{\mathrm{rc}}, z_{l,t}^{\mathrm{rc}})}{\partial z_{w,t}} = -\alpha_t \frac{\partial \mathcal{L}}{\partial z_{w,t}^{\mathrm{rc}}} = -\alpha_t \frac{\partial \mathcal{L}}{\partial z_{w,t}} = \alpha_t d_{w,t},$$

$$d_{l,t}^{\mathrm{rc}} = \frac{\partial \mathcal{L}(z_{w,t}^{\mathrm{rc}}, z_{l,t}^{\mathrm{rc}})}{\partial z_{l,t}} = \alpha_t^{-1} \frac{\partial \mathcal{L}}{\partial z_{l,t}^{\mathrm{rc}}} = \alpha_t^{-1} \frac{\partial \mathcal{L}}{\partial z_{l,t}} = \alpha_t^{-1} d_{l,t}. \tag{60}$$

Finally, letting $r_t^{\mathrm{rc}} \triangleq d_{w,t}^{\mathrm{rc}}/d_{l,t}^{\mathrm{rc}}$ and using Eq. (60), we have

$$r_t^{\mathrm{rc}} = \frac{\alpha_t d_{w,t}}{\alpha_t^{-1} d_{l,t}} = \alpha_t^2 \frac{d_{w,t}}{d_{l,t}} = \alpha_t^2 r_t.$$

Taking logs gives $\log r_t^{\mathrm{rc}} = \log r_t + 2 \log \alpha_t$. By the definition of $\alpha_t$, $2 \log \alpha_t = \log r_t^\star - \log r_t$, hence

$$\log r_t^{\mathrm{rc}} = \log r_t^\star,$$

which completes the proof.

$\square$

## C. Experimental Settings and Results

### C.1. Dataset Details

**UltraFeedback Binarized** (Cui et al., 2024; Tunstall et al., 2024). We evaluate our method on the UltraFeedback Binarized dataset, a widely adopted benchmark comprising approximately 64k prompts derived from the UltraFeedback compilation. The dataset constructs binary preference pairs $(\boldsymbol{x}, \boldsymbol{y}_w, \boldsymbol{y}_l)$ by leveraging scalar quality scores assigned by GPT-4 across multiple dimensions (e.g., helpfulness, honesty) for four distinct model completions. Specifically, the completion achieving the highest aggregate score is designated as the chosen response $\boldsymbol{y}_w$, while the rejected response $\boldsymbol{y}_l$ is stochastically sampled from the remaining candidates. This process effectively converts fine-grained scalar signals into the binary contrastive pairs required for preference optimization.

### C.2. Evaluation Protocols and Task Descriptions

To comprehensively assess whether our reward calibration method improves alignment without compromising general capabilities, we employ a diverse set of benchmarks covering reasoning, mathematics, and general knowledge. Specifically, models fine-tuned on the UltraFeedback Binarized dataset are evaluated on the HuggingFace Open LLM Leaderboard (v1 and v2)[2] (Beeching et al., 2023; Fourrier et al., 2024).

Below, we detail the specific tasks included in our quantitative evaluation:

- **MMLU-Pro (Wang et al., 2024b).** An advanced iteration of the Massive Multitask Language Understanding benchmark. MMLU-Pro introduces more complex multiple-choice questions and undergoes rigorous expert review to enhance difficulty and reduce data biases. It serves as a robust indicator of the model's broad knowledge and reasoning stability.

- **BBH (Big Bench Hard) (Suzgun et al., 2022).** A subset of 23 challenging tasks selected from the BIG Bench suite, designed to test capabilities where standard language models typically struggle. It focuses on symbolic reasoning, reading comprehension, and logical deduction.

- **MUSR (Multi-step Soft Reasoning) (Sprague et al., 2024).** A benchmark designed to evaluate the limits of chain-of-thought reasoning. It presents complex scenarios that require the model to integrate information and reason across long contexts to derive correct conclusions.

- **MATH (Hendrycks et al., 2021).** A compilation of challenging mathematics problems derived from high-school competitions. The dataset utilizes LaTeX and Asymptote formatting to ensure precision, serving as a rigorous test of the model's advanced mathematical problem-solving skills.

- **GSM8k (Cobbe et al., 2021).** A canonical benchmark consisting of high-quality grade school math problems. We employ a 5-shot setting to evaluate the model's capability to perform multi-step mathematical reasoning and navigate linguistic complexity in problem statements.

- **ARC (AI2 Reasoning Challenge) (Clark et al., 2018).** Focuses on grade school level science questions that require factual knowledge retrieval and logical reasoning. We utilize the 25-shot evaluation protocol to assess the model's ability to generalize scientific concepts to novel questions.

### C.3. Implementation Details of Reward Calibration

In this section, we detail the algorithm used to estimate the target ratio $r_t^\star$ and the realized ratio $r_t$.

**Gradient Approximation via the Output Layer.** Calculating the exact score norms $\|\boldsymbol{s}_{w|l,t}\| = \|\nabla_{\boldsymbol{\theta}} z_{w|l}(\boldsymbol{\theta}_t)\|$ involves a full backward pass through the model. To maintain training throughput, we approximate these norms using a subset of parameters $\boldsymbol{\theta}_{\text{head}}$ closest to the output. Specifically, our implementation selects parameters based on the training mode:

- **Full Fine-Tuning**: We use the weights of the language model head (the final linear projection to the vocabulary).

---

[2] https://huggingface.co/spaces/open-llm-leaderboard/open_llm_leaderboard

- **Parameter-Efficient Fine-Tuning (e.g., LoRA (Hu et al., 2022) or QLoRA (Dettmers et al., 2023))**: Since the head is typically frozen, we use the trainable parameters of the final LoRA adapter layers: $\|\boldsymbol{s}_{w|l,t}\| \approx \|\nabla_{\boldsymbol{\theta}_{\text{head}}} z_{w|l}(\boldsymbol{\theta}_t)\|$.

To validate this approximation, we compare the DB computed using full-parameter gradients versus head-only gradients on Mistral-7B. As shown in Fig. 6, the resulting bands overlap significantly for both DPO and DIL-BCE. This suggests that the main geometric information required for calibration ($\rho_t$ and norm ratios $\|\boldsymbol{s}_{l,t}\|/\|\boldsymbol{s}_{w,t}\|$) is captured sufficiently well for calibration.

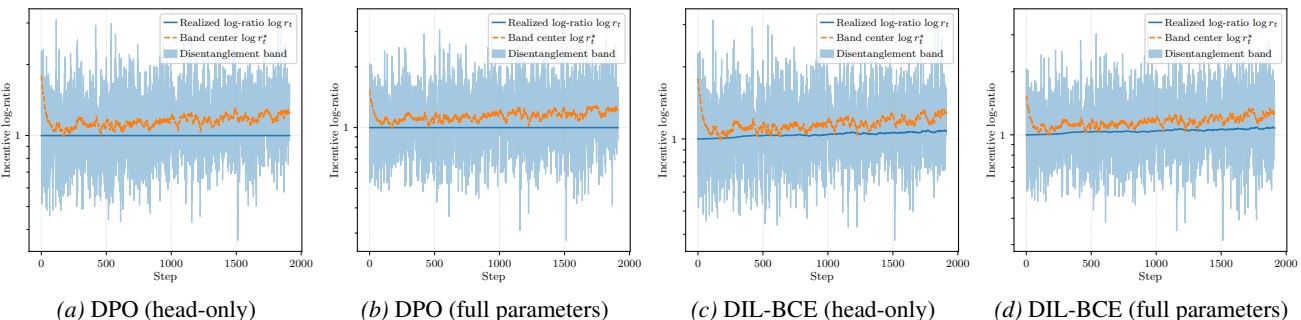

*(a)* DPO (head-only)    *(b)* DPO (full parameters)    *(c)* DIL-BCE (head-only)    *(d)* DIL-BCE (full parameters)

*Figure 6.* Validation of the head-only gradient approximation on Mistral-7B. We compare the DB computed using gradients from only the output layer (head-only) versus all trainable parameters (full parameters). For both entangled (DPO) and disentangled (DIL-BCE) objectives, the head-only approximation closely matches the width and trend of the DB, justifying its use for efficient calibration.

**Log-Domain EMA.**    The incentive coefficients and gradient norms fluctuate due to mini-batch noise and optimization dynamics. To estimate a robust center, we maintain exponential moving averages (EMAs) of their logarithms. Specifically, for the incentive coefficients $\log d_{w,t}$ and $\log d_{l,t}$, we update:

$$
\begin{aligned}
\text{EMA}[\log d_{w,t}] &= \beta \cdot \text{EMA}[\log d_{w,t-1}] + (1-\beta) \cdot \log(d_{w,t} + \epsilon), \\
\text{EMA}[\log d_{l,t}] &= \beta \cdot \text{EMA}[\log d_{l,t-1}] + (1-\beta) \cdot \log(d_{l,t} + \epsilon),
\end{aligned}
\tag{61}
$$

where $\beta$ is the momentum parameter. The same update rule is applied to $\|\mathbf{s}_{w,t}\|$ and $\|\mathbf{s}_{l,t}\|$ to obtain $\text{EMA}[\log \|\boldsymbol{s}_{w,t}\|]$ and $\text{EMA}[\log \|\boldsymbol{s}_{l,t}\|]$.

The smoothed log-ratios are then computed as:

$$
\begin{aligned}
\text{EMA}[\log r_t] &= \text{EMA}[\log d_{w,t}] - \text{EMA}[\log d_{l,t}], \\
\text{EMA}[\log r_t^\star] &= \text{EMA}[\log \|\mathbf{s}_{l,t}\|] - \text{EMA}[\log \|\mathbf{s}_{w,t}\|].
\end{aligned}
\tag{62}
$$

This approach smooths via the geometric mean, which is more robust to scale variations than the arithmetic mean.

**Dynamic Clipping.**    To ensure the calibration factor

$$
\left( \frac{\exp(\text{EMA}[\log r_t^\star])}{\exp(\text{EMA}[\log r_t])} \right)^{1/2} = \exp\left( \frac{1}{2}(\text{EMA}[\log r_t^\star] - \text{EMA}[\log r_t]) \right),
\tag{63}
$$

respects the feasible region derived in Eq. (9), we clip its log-domain value based on the instantaneous cosine similarity $\rho_t$.

Specifically, we constrain the log-calibration term $\text{EMA}[\log r_t^\star] - \text{EMA}[\log r_t]$ to the interval:

$$
\left[ \text{EMA}[\log r_t^\star] - \log\frac{d_{w,t}}{d_{l,t}} + \log \rho_t, \, \text{EMA}[\log r_t^\star] - \log\frac{d_{w,t}}{d_{l,t}} - \log \rho_t \right].
\tag{64}
$$

This guarantees that the calibrated ratio always lies within the DB bounds of the current batch.

**Multi-Epoch Training.**    For longer training, a multi-epoch ablation on Mistral-7B with DPO suggests that RC does not degrade performance: at 2 epochs, results are similar ($36.84 \rightarrow 36.93$), while at 1 epoch RC shows a larger effect ($24.26 \rightarrow 36.44$). This indicates RC may be more helpful when training is unstable and appears non-disruptive under longer runs.

*Table 7.* Ablation study on training epochs for RC under DPO on Mistral-7B. Subscripts denote standard errors.

| Method | Epoch | ARC | BBH | MuSR | MATH | GSM8K | MMLU-Pro | Average |
|---|---|---|---|---|---|---|---|---|
| Baseline | | $60.32_{1.43}$ | $44.68_{.61}$ | $41.93_{1.74}$ | $2.92_{.47}$ | $36.69_{1.33}$ | $30.23_{.42}$ | 36.13 |
| w/o RC | 1 | $22.70_{1.22}$ | $29.12_{2.65}$ | $41.53_{1.74}$ | $2.93_{.46}$ | $37.60_{1.33}$ | $11.66_{.29}$ | 24.26 |
| | 2 | $62.12_{1.42}$ | $44.67_{.61}$ | $41.53_{1.74}$ | $2.57_{.44}$ | $39.73_{1.35}$ | $30.39_{.42}$ | 36.84 |
| w/ RC | 1 | $60.92_{1.43}$ | $44.83_{3.01}$ | $41.67_{1.74}$ | $2.70_{.45}$ | $38.21_{1.34}$ | $30.31_{.42}$ | 36.44 |
| | 2 | $62.20_{1.42}$ | $44.94_{.61}$ | $41.53_{1.75}$ | $2.64_{.44}$ | $39.88_{1.35}$ | $30.39_{.42}$ | 36.93 |

**Runtime Overhead of Reward Calibration.** Table 8 summarizes the estimated end-to-end training duration w/o RC and w/ RC. Overhead is small in practice: for LoRA fine-tuning on Mistral-7B and Pythia-2.8B, differences are minor and occasionally negative, consistent with normal system/runtime variance. A more visible relative overhead appears for full-parameter fine-tuning on Pythia-410M, where the backward pass is lightweight and frequent gradient-norm computations are less amortized. Overall, RC remains a plug-and-play dynamics intervention with minimal additional wall-clock cost in the main experimental regimes.

*Table 8.* Estimated Training Duration (w/o RC vs. w/ RC). Projected total wall-clock time from stable tqdm ETA for each model and objective; the table reports times w/o RC and w/ RC, together with absolute and relative differences.

| Model | Metric | BCE | CPO | DDRO | DPO | IPO | LSIF | SimPO | UKL | Average |
|---|---|---|---|---|---|---|---|---|---|---|
| **Mistral-7B** | w/o RC Time | 10h 46m | 11h 00m | 11h 52m | 10h 51m | 11h 01m | 10h 46m | 10h 44m | 11h 00m | 11h 00m |
| | w/ RC Time | 11h 14m | 11h 19m | 11h 30m | 11h 15m | 11h 18m | 11h 14m | 11h 45m | 11h 19m | 11h 22m |
| | Δ Time | +28m | +19m | −22m | +25m | +17m | +29m | +1h 01m | +19m | +22m |
| | Δ (%) | +4.3% | +2.9% | −3.0% | +3.8% | +2.6% | +4.4% | +9.4% | +2.9% | +3.3% |
| **Pythia-2.8B** | w/o RC Time | 4h 34m | 4h 30m | 4h 32m | 4h 19m | 4h 30m | 4h 19m | 4h 32m | 4h 32m | 4h 29m |
| | w/ RC Time | 4h 24m | 4h 30m | 4h 21m | 4h 25m | 4h 30m | 4h 21m | 4h 21m | 4h 19m | 4h 24m |
| | Δ Time | −10m | −0m | −12m | +06m | −0m | +02m | −11m | −13m | −05m |
| | Δ (%) | −3.6% | −0.1% | −4.3% | +2.3% | −0.1% | +0.6% | −4.1% | −4.7% | −1.8% |
| **Pythia-410M** | w/o RC Time | 48m | 53m | 43m | 46m | 52m | 53m | 50m | 52m | 50m |
| | w/ RC Time | 1h 00m | 54m | 59m | 1h 02m | 58m | 1h 00m | 58m | 1h 00m | 59m |
| | Δ Time | +12m | +0m | +16m | +16m | +06m | +07m | +08m | +08m | +09m |
| | Δ (%) | +24.9% | +0.6% | +36.5% | +34.8% | +11.4% | +13.5% | +16.1% | +15.0% | +18.4% |

## C.4. Additional Results of Downstream Performance Comparison On Benchmarks

Tabs. 4 and 9 report downstream benchmark scores w/ and w/o RC.

*Table 9.* Evaluation results on benchmarks across objectives using Pythia-2.8B. Subscripts denote standard errors.

| Method | RC | MMLU-Pro | BBH | MUSR | ARC | Average ($\Delta$) |
|---|---|---|---|---|---|---|
| DPO | w/o | $10.82_{.28}$ | $31.83_{.58}$ | $30.95_{1.61}$ | $34.04_{1.38}$ | 26.91 |
| | w/ | $11.27_{.29}$ | $32.31_{.58}$ | $34.39_{1.69}$ | $37.12_{1.41}$ | 28.77 (+1.86) |
| SimPO | w/o | $10.94_{.28}$ | $30.49_{.57}$ | $32.14_{1.65}$ | $32.59_{1.37}$ | 26.54 |
| | w/ | $11.54_{.29}$ | $31.34_{.57}$ | $35.45_{1.68}$ | $37.03_{1.41}$ | 28.84 (+2.30) |
| CPO | w/o | $11.78_{.29}$ | $31.70_{.58}$ | $35.32_{1.71}$ | $37.88_{1.42}$ | 29.17 |
| | w/ | $11.11_{.29}$ | $32.22_{.58}$ | $35.71_{1.71}$ | $39.08_{1.43}$ | 29.53 (+0.36) |
| BCE | w/o | $10.99_{.29}$ | $31.36_{.58}$ | $34.26_{1.67}$ | $36.60_{1.41}$ | 28.30 |
| | w/ | $11.54_{.29}$ | $32.12_{.58}$ | $33.86_{1.67}$ | $37.37_{1.41}$ | 28.72 (+0.42) |
| DDRO | w/o | $12.08_{.30}$ | $31.62_{.58}$ | $36.51_{1.71}$ | $38.31_{1.42}$ | 29.63 |
| | w/ | $11.34_{.29}$ | $32.40_{.59}$ | $32.67_{1.67}$ | $36.69_{1.41}$ | 28.28 (−1.35) |
| LSIF | w/o | $11.29_{.29}$ | $31.62_{.58}$ | $31.22_{1.64}$ | $35.32_{1.40}$ | 27.36 |
| | w/ | $11.30_{.29}$ | $32.96_{.59}$ | $33.33_{1.67}$ | $37.54_{1.42}$ | 28.78 (+1.42) |
| TI-DPO | w/o | $10.89_{.28}$ | $32.12_{2.85}$ | $30.95_{1.63}$ | $34.56_{1.39}$ | 27.13 |
| | w/ | $11.30_{.29}$ | $32.19_{2.83}$ | $32.80_{1.65}$ | $37.46_{1.41}$ | 28.44 (+1.31) |
| IPO | w/o | $11.03_{.29}$ | $30.64_{.56}$ | $31.08_{2.78}$ | $34.04_{1.38}$ | 26.70 |
| | w/ | $11.20_{.29}$ | $31.53_{.58}$ | $35.05_{2.91}$ | $38.14_{1.42}$ | 28.98 (+2.28) |
| UKL | w/o | $10.84_{.28}$ | $30.27_{.58}$ | $35.05_{2.93}$ | $37.03_{1.41}$ | 28.30 |
| | w/ | $11.32_{.29}$ | $30.91_{.57}$ | $34.66_{2.90}$ | $37.29_{1.41}$ | 28.55 (+0.25) |

---

**Algorithm 2** PyTorch Implementation of Reward Calibration.

---

```python
def ratio_calibrated_loss(model, batch, base_loss_fn):
    # 1. Forward pass to get rewards
    # zw, zl: shape (B,), requires_grad=True
    zw, zl = model(batch['chosen']), model(batch['rejected'])

    # 2. Compute score norms and incentive coefficients
    # Using head-only gradients for efficiency, as discussed in Sec. C.3
    head_params = model.lm_head.parameters()
    sw_norm = torch.autograd.grad(zw.mean(), head_params, retain_graph=True)[0].norm()
    sl_norm = torch.autograd.grad(zl.mean(), head_params, retain_graph=True)[0].norm()

    # Evaluate incentive coefficients in Tab. 1
    dw, dl = get_incentive_coefficients(model, batch, base_loss_fn, zw, zl)

    # 3. Compute Calibration Factor
    # Note: In practice, we use log-space EMA for stability, as discussed in Sec. C.3
    # Evaluate target ratio r* = sl / sw discussed in Proposition 3.3
    r_star = sl_norm / sw_norm.clamp(min=1e-8)

    # Current ratio r is implicitly handled by the backward scaling
    r = dw / dl.clamp(min=1e-8)

    # 4. Construct Calibrated Rewards via  Eq. (12)
    # Forward: unchanged; Backward: gradients scaled by alpha
    alpha = torch.sqrt(r_star / r)
    zw_rc = alpha * zw + (1 - alpha) * zw.detach()
    zl_rc = (1 / alpha) * zl + (1 - 1 / alpha) * zl.detach()

    # 5. Compute Objective
    return base_loss_fn(zw_rc, zl_rc)

```

## C.5. Additional Results on Preference Optimization Dynamics w/ or w/o RC

In this section, we present an exhaustive set of learning dynamics analyses for all evaluated objectives across four model scales: Pythia-410M, Pythia-2.8B, Mistral-7B, and Qwen2.5-7B. These visualizations provide evidence for our DB-based diagnosis: when training-time updates keep the realized log-ratio $\log r_t$ largely inside the DB and away from its boundaries, the trajectories tend to concentrate on the Pathway (iii): chosen not down, rejected suppressed. RC offers a simple mechanism that often moves the dynamics toward this regime by adaptively rebalancing the chosen vs. rejected updates.

**Detailed Analysis of the Preference Optimization Dynamics on Pythia-410M.** Sec. C.5 visualizes, for each method, three coupled signals over steps: (1) whether $\log r_t$ lies inside DB, (2) likelihood trajectory, and (3) margin evolution. Across methods, the observed likelihood pathways are largely consistent with the ratio axis regimes summarized in Tab. 2.

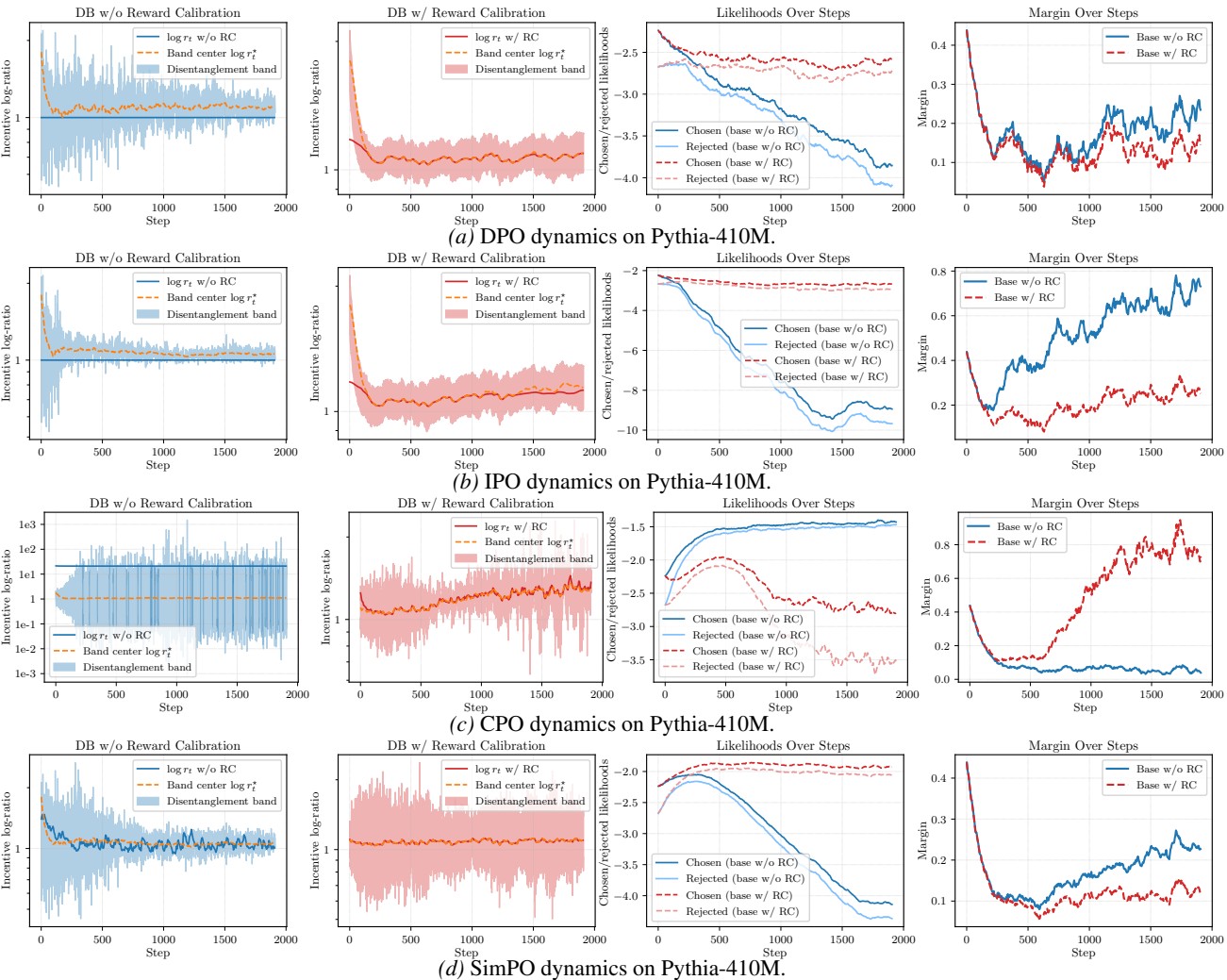

*(a)* DPO dynamics on Pythia-410M.

*(b)* IPO dynamics on Pythia-410M.

*(c)* CPO dynamics on Pythia-410M.

*(d)* SimPO dynamics on Pythia-410M.

*Figure 7.* Preference optimization dynamics of Pythia-410M under entangled margin-based objectives.

In the standard baseline runs ("Base w/o RC"), DB violations often coincide with Pathway (i)/(ii). For DPO and IPO, $\log r_t$ repeatedly dips below the *lower* DB boundary (Figs. 7a and 7b). As predicted, this coincides with Pathway (ii): both $z_{w,t}$ and $z_{l,t}$ decrease, indicating that suppressing the loser comes with an unintended degradation of the winner. In contrast, CPO often pushes $\log r_t$ toward (or beyond) the upper DB boundary (Fig. 7c), yielding Pathway (i) where both likelihoods rise together, i.e., the update is insufficiently selective to suppress $z_{l,t}$ alone. SimPO typically stays closer to DB but can still drift toward the boundaries depending on steps, leading to mixed behaviors (Fig. 7d).

Upon applying RC ("Base w/ RC"), ratio centering often promotes the Pathway (iii)-like regime. After applying RC, $\log r_t$ is typically anchored closer to $\log r_t^\star$ and remains inside DB for a larger fraction of training across objectives. Correspondingly, the likelihood trajectories more frequently exhibit the Pathway (iii) signature: $z_{w,t}$ is not down (often mildly increasing) while $z_{l,t}$ is suppressed. Notably, RC is not limited to entangled margin baselines: for density ratio objectives (e.g., DIL-UKL and DDRO), RC can also reduce boundary-hugging and make the desired regime more persistent (Sec. C.5). Overall, Pythia-410M provides a clean controlled setting where the association between DB feasibility and the induced pathway is visually clearest.

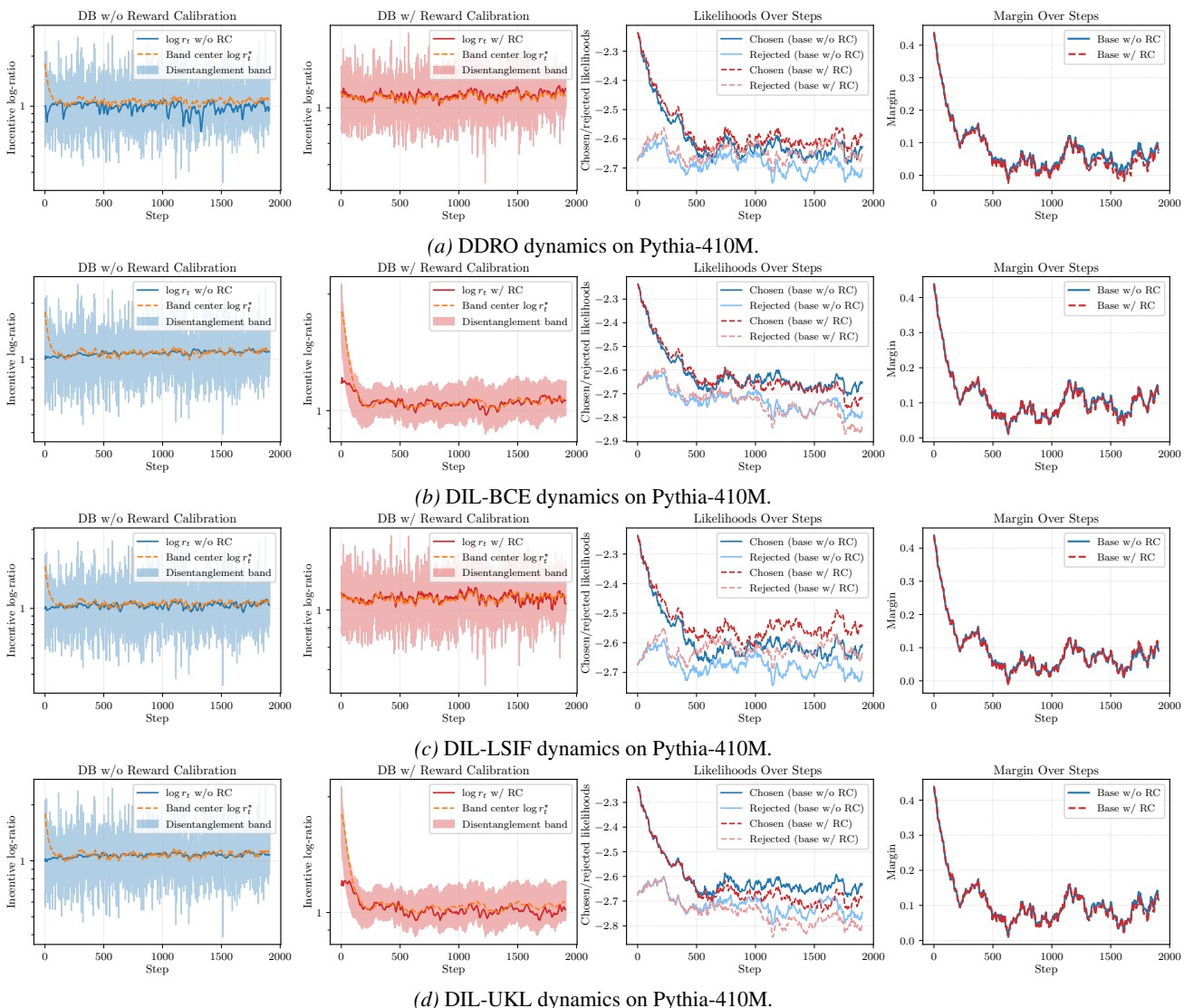

*(a)* DDRO dynamics on Pythia-410M.

*(b)* DIL-BCE dynamics on Pythia-410M.

*(c)* DIL-LSIF dynamics on Pythia-410M.

*(d)* DIL-UKL dynamics on Pythia-410M.

*Figure 8.* Preference optimization dynamics of Pythia-410M under disentangled density ratio-based objectives.

**Detailed Analysis of the Preference Optimization Dynamics on Pythia-2.8B.** Scaling to Pythia-2.8B strengthens (rather than weakens) the separation between feasible and infeasible ratio regimes. The same DB-based diagnosis continues to explain the observed pathways, but ratio excursions translate into more pronounced likelihood drifts.

For Entangled baselines (w/o RC), we observe that there are amplified Pathway (ii) and Pathway (i) failures. DPO, IPO, SimPO and TI-DPO exhibit frequent excursions to the lower DB side (Figs. 9a to 9d), producing a clearer Pathway (ii) pattern than at 410M: $z_{w,t}$ degrades alongside $z_{l,t}$ over substantial training intervals. CPO, on the other hand, can move

$\log r_t$ toward the upper DB boundary (Fig. 9e), consistent with Pathway (i) where both likelihoods increase and loser suppression becomes less selective.

More importantly, a "disentangled form" alone does not guarantee Pathway (iii). While density ratio-based objectives often reduce lockstep coupling, they are not automatically DB-feasible under a given model/data/optimizer configuration. DDRO on Pythia-2.8B provides a concrete example: its baseline $\log r_t$ can lie visibly outside DB (Fig. 10a), and the corresponding likelihood evolution deviates from Pathway (iii). This supports our central premise: the realized training-time balance captured by DB feasibility is a more general predictor of the pathway than the high-level objective family label.

Upon applying RC, Pathway (iii) is often recovered across objectives in our evaluated settings. RC centers $\log r_t$ closer to $\log r_t^\star$ and keeps it within DB for a larger fraction of steps across objectives. The likelihood trajectories then more frequently exhibit the desired profile: the chosen likelihood is preserved (not down overall) while the rejected likelihood is suppressed (see Fig. 4 and Sec. C.5). In terms of margins, RC typically maintains steady margin growth, though its magnitude can differ from the uncalibrated baseline, reflecting a reallocation of update emphasis toward DB-feasible suppression.

**Detailed Analysis of the Preference Optimization Dynamics on Mistral-7B.** Mistral-7B largely follows the same DB logic. For most objectives, RC improves ratio feasibility and promotes Pathway (iii). For DPO and the DIL-BCE family, RC visibly pulls $\log r_t$ toward $\log r_t^\star$ and stabilizes the chosen likelihood compared to the baseline (Figs. 11a and 12b). Similar improvements appear for SimPO, DDRO, and LSIF, where RC reduces boundary-hugging behavior and yields trajectories closer to Pathway (iii) (Figs. 11d, 12a and 12c).

For CPO on Mistral-7B (Fig. 11c), RC still centers the realized log-ratio near $\log r_t^\star$ and keeps it largely within DB. In this setting, the likelihood traces can exhibit short intervals where both likelihoods move downward together, which is consistent with our regime-based interpretation: DB characterizes an instantaneous feasibility condition, and the desired Pathway (iii) behavior is best understood as a training-time regime that the trajectory can enter and revisit (often after an early stage), rather than a guarantee of monotone per-step drift over the entire run. Empirically, with RC the trajectory spends substantially more of training in the DB-feasible region and shows a more Pathway (iii)-like profile overall, compared to the uncalibrated baseline.

Across Pythia-410M, Pythia-2.8B, and Mistral-7B, we observe a consistent empirical pattern: when $\log r_t$ stays inside DB and remains away from its boundaries for substantial portions of training, the dynamics tend to exhibit Pathway (iii)-like behavior; when $\log r_t$ persistently violates DB, the trajectories more often resemble Pathway (i) or Pathway (ii). RC provides a simple and generally effective mechanism to improve this DB feasibility by rebalancing chosen vs. rejected updates, making it more likely for training to enter and stay in the desired Pathway (iii) regime.

**Detailed Analysis of the Preference Optimization Dynamics on Qwen2.5-7B.** Qwen2.5-7B shows the same DB-based picture, but in a milder form than Mistral-7B. In Fig. 13, for the entangled DPO objective, the uncalibrated run shows visible fluctuation around the DB and a gradual downward drift of the chosen likelihood. After applying RC, the realized log-ratio is pulled closer to the band center, the likelihood trajectories become more stable, and the margin is still preserved.

In Fig. 14, for the disentangled DIL-BCE objective, the baseline is already relatively stable, so the effect of RC is smaller. Even so, RC still keeps the realized log-ratio closer to the DB center and slightly reduces likelihood drift, while leaving the margin trajectory largely unchanged.

Overall, the Qwen2.5-7B results suggest that when the base dynamics are already fairly stable, RC is mostly non-disruptive, while still helping training stay closer to a Pathway (iii)-like regime.

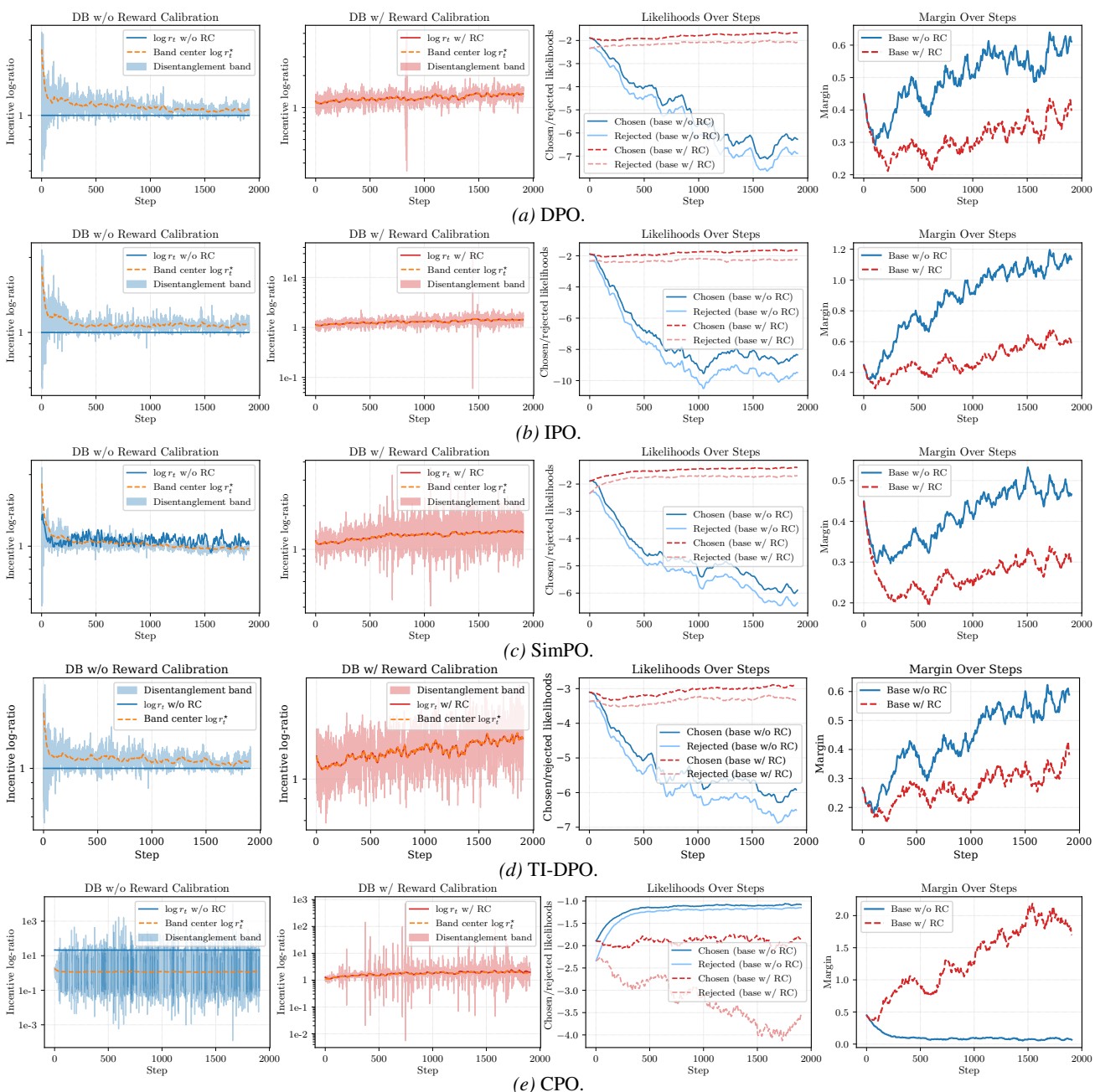

*Figure 9.* Preference optimization dynamics of Pythia-2.8B under entangled margin-based objectives.

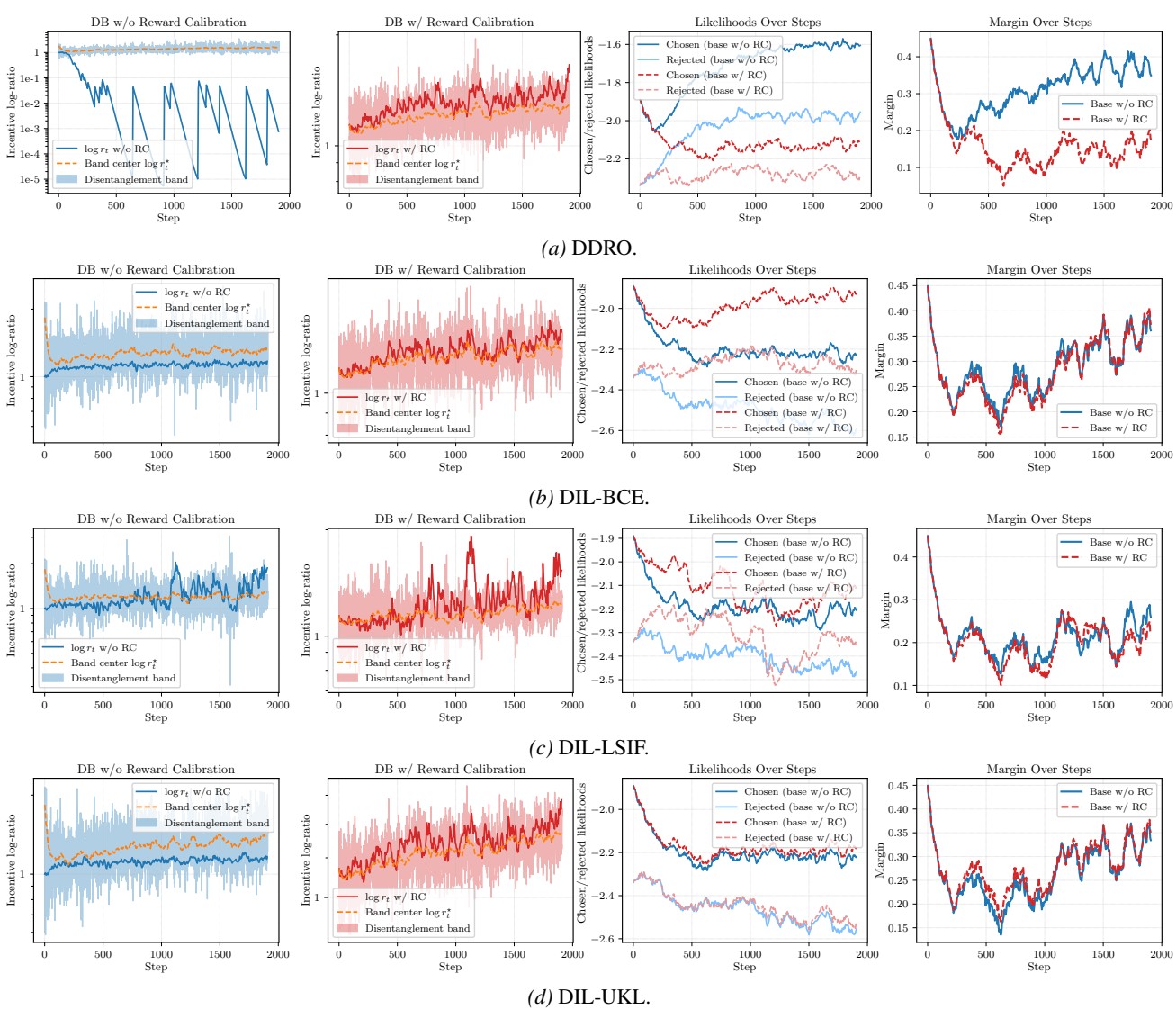

*Figure 10.* Preference optimization dynamics of Pythia-2.8B under disentangled density ratio-based objectives.

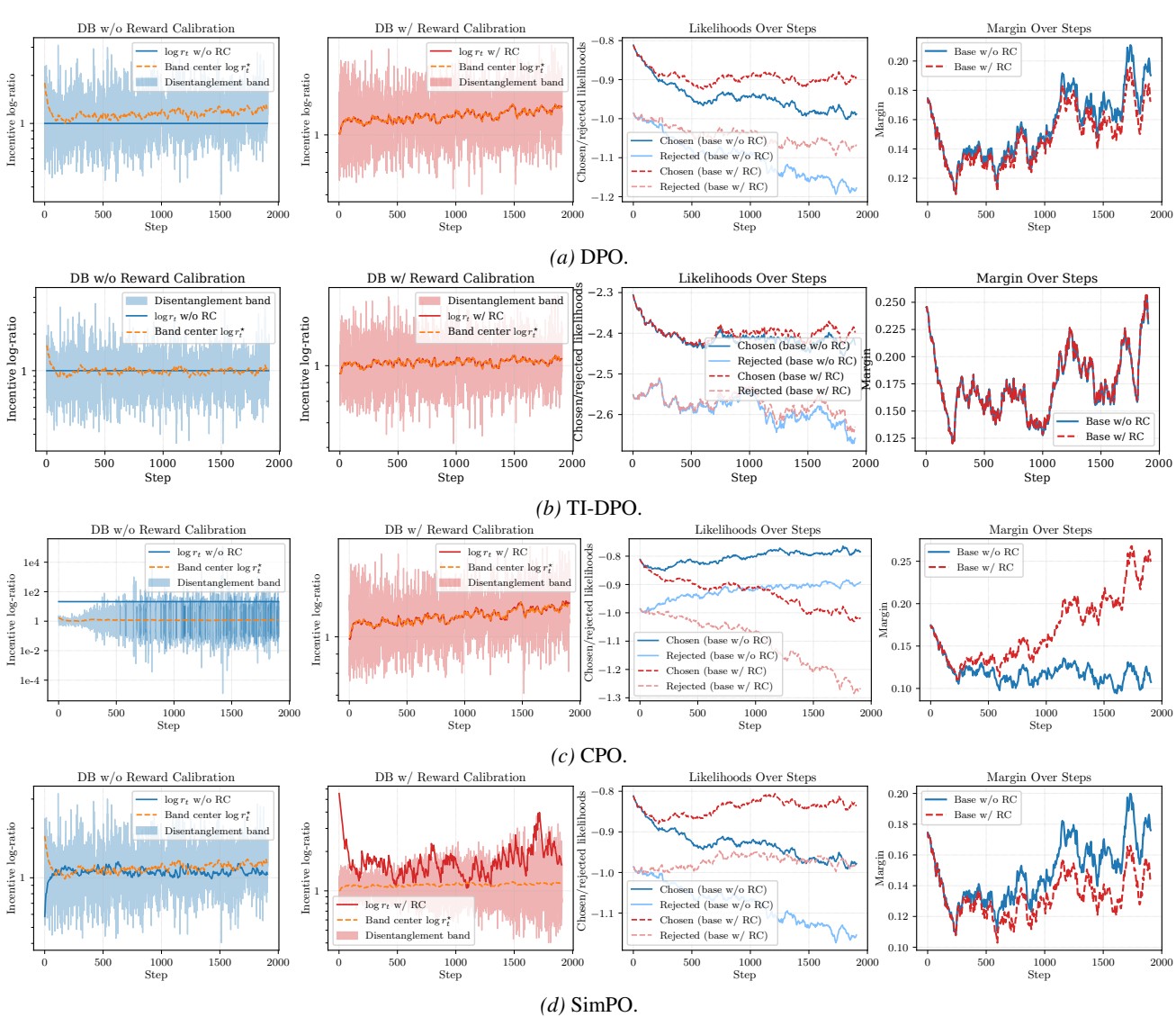

*Figure 11.* Preference optimization dynamics of Mistral-7B under entangled objectives.

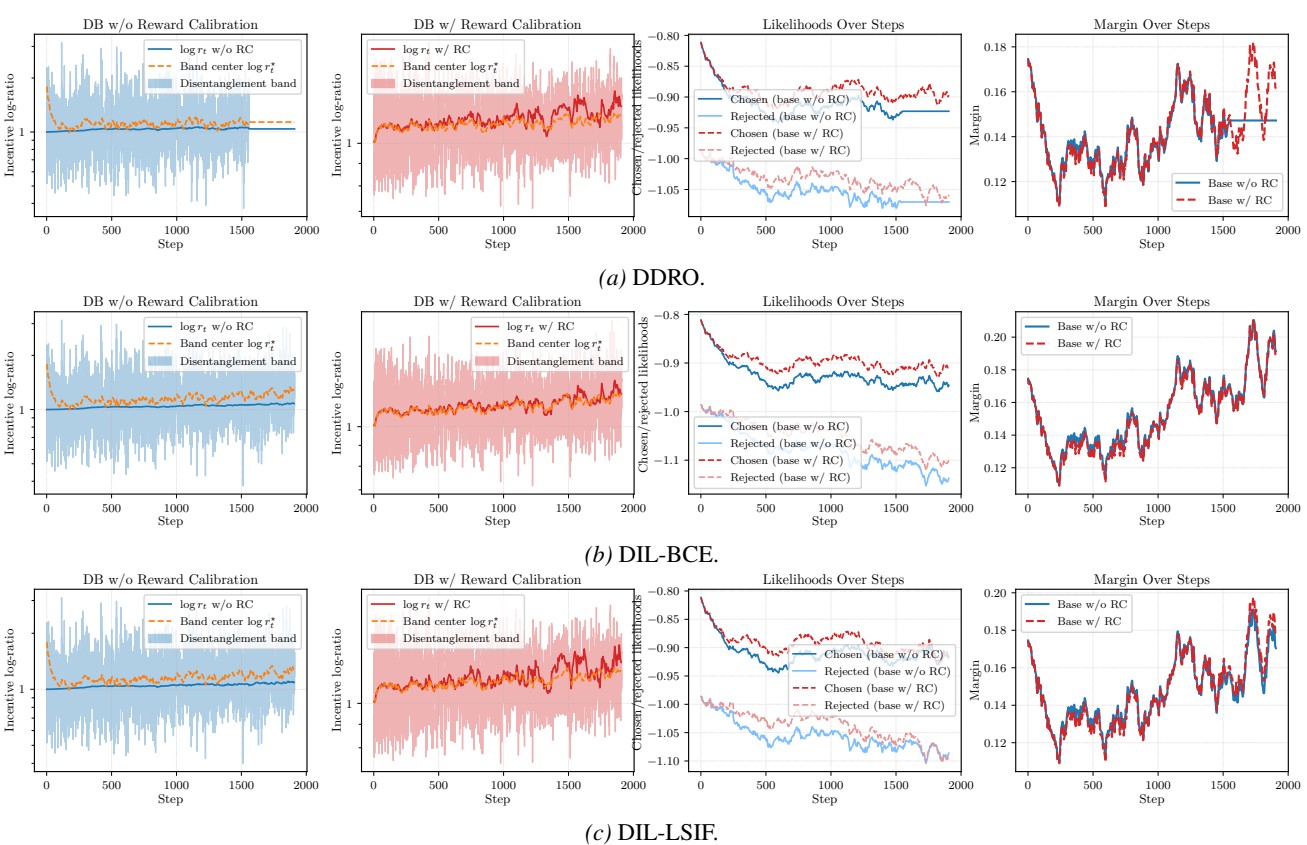

*Figure 12.* Preference optimization dynamics of Mistral-7B under disentangled objectives.

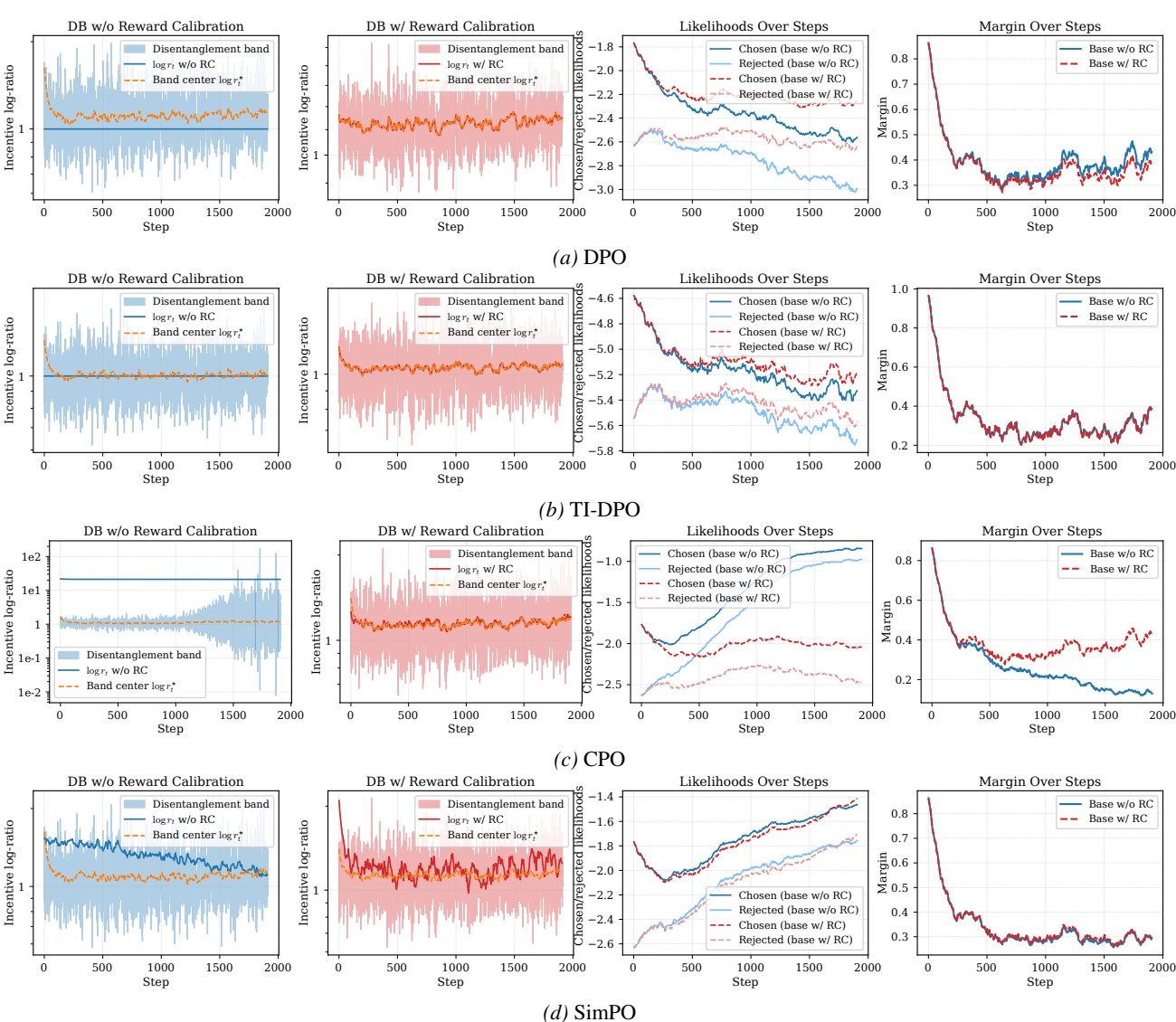

*Figure 13.* Preference optimization dynamics of Qwen-7B under entangled objectives.

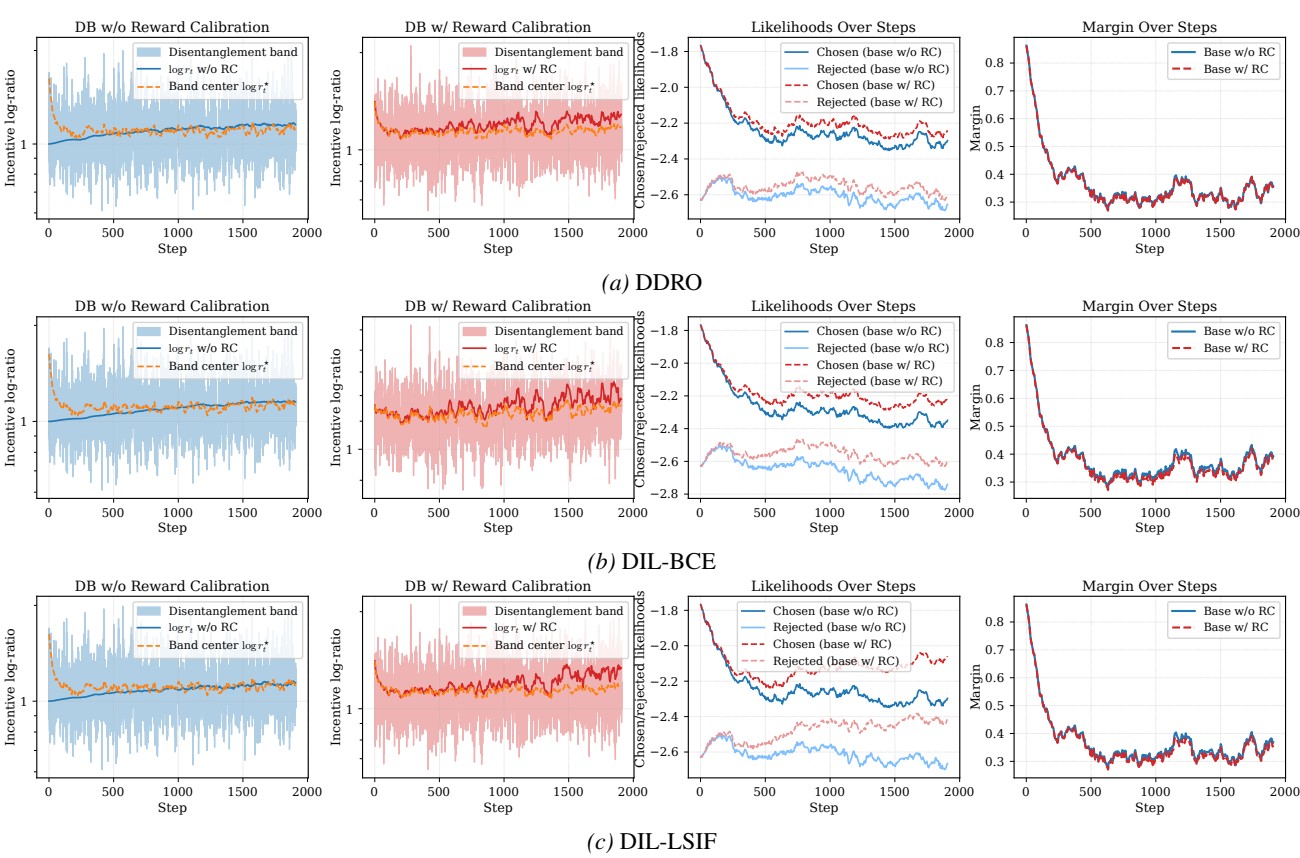

*Figure 14.* Preference optimization dynamics of Qwen-7B under disentangled objectives.

