# OpenReview forum: "Towards Disentangled Preference Optimization Dynamics: Suppress the Loser, Preserve the Winner"
_ICML.cc/2026/Conference — ICML 2026 regular_

### Official Review · Reviewer_JgeZ · 2026-02-21

**Soundness:** 3
**Presentation:** 4
**Significance:** 2
**Originality:** 3
**Overall Recommendation:** 4
**Confidence:** 4

**Summary:**

This paper studies the optimization dynamics of preference-based fine-tuning for aligning large language models with human preferences. The authors present a unified incentive-score gradient decomposition, showing that many existing preference objectives share identical update directions and differ only in scalar incentive coefficients. Based on a continuous-time analysis of reward dynamics, the paper introduces the Disentanglement Band (DB), a testable interval characterizing when training can suppress the rejected response without degrading the chosen one (Pathway iii). Leveraging this analysis, the authors propose Reward Calibration (RC), a lightweight, plug-and-play backward-pass wrapper that rescales chosen and rejected gradients to keep the incentive ratio near the DB center. Experiments on UltraFeedback with Pythia-410M, Pythia-2.8B, and Mistral-7B show that RC stabilizes training dynamics across a range of preference objectives and often improves downstream performance, especially on reasoning benchmarks.

**Compliance With Llm Reviewing Policy:**

Affirmed.

**Final Justification:**

All my concerns have been adequately addressed. I have raised my overview score to a weak accept.

**Key Questions For Authors:**

1.	How sensitive is Reward Calibration to the EMA momentum and batch size used to estimate incentive ratios?
2.	Why are all experiments restricted to one epoch of training? Does RC help or hurt longer training runs?
3.	Are there failure cases where RC misestimates the DB due to negative or unstable cosine similarity?
4.	How would RC extend to settings with more than two responses per prompt (e.g., multi-winner or ranking-based datasets)?

**Limitations:**

The paper evaluates RC only on a single preference dataset and a limited set of benchmarks, without reporting variance across random seeds. Improvements are sometimes driven by recovering from unstable baseline runs, making it difficult to assess consistent gains. The approximation of gradient norms using head-only or adapter gradients may also misestimate the DB in some settings. Broader empirical validation and ablations are needed.

**Strengths And Weaknesses:**

**Strengths:**
- The incentive-score decomposition provides a clear and unifying perspective on a wide family of preference optimization objectives.
- The Disentanglement Band is intuitive, easy to verify, and gives a concrete geometric criterion for understanding winner–loser reward dynamics.
- Reward Calibration is simple, objective-agnostic, and computationally lightweight, with minimal wall-clock overhead in LoRA settings.
- The paper is generally well written and clearly structured.

**Weaknesses:**
- Empirical evaluation is limited to a single preference dataset (UltraFeedback) and one-epoch training, raising concerns about generalization. I want to know whether RC is a boon or a hindrance for longer training cycles?
- Several downstream improvements are modest and sometimes negative; claims of “consistent improvement” are overstated.
- Baseline comparisons omit recent multi-dimensional or structured preference optimization methods, such as TDPO [1] and TI-DPO [2]
- Ablation studies on RC design choices (EMA momentum, gradient approximation, calibration strength) are missing.


[1] Zeng Y, Liu G, Ma W, et al. Token-level direct preference optimization[J]. arXiv preprint arXiv:2404.11999, 2024.

[2] Yang N, Lin H, Liu Y, et al. Token-Importance Guided Direct Preference Optimization[J]. arXiv preprint arXiv:2505.19653, 2025.

---

> ### Author Rebuttal · Authors · 2026-03-30
>
> Thank you for the careful and constructive review. We appreciate the reviewer’s positive assessment of the unified incentive-score perspective, the DB-based criterion, and the lightweight RC design, and we address the main concerns point by point below.
>
> > Q1+W4+L3：Ablation studies on RC design choices (EMA momentum, gradient approximation, calibration strength) are missing.
>
> (1) EMA momentum. We add an EMA ablation on Pythia-2.8B with DPO. Across momenta (0.5, 0.9, 0.98, 0.999), results are similar, and RC outperforms w/o RC on average, suggesting limited sensitivity to EMA momentum.
>
> |EMA|MMLU-PRO|BBH|MUSR|ARC|Avg|
> |---|---|---|---|---|---|
> |w/o RC|10.82±0.28|31.83±0.58|30.95±1.61|34.04±1.38|26.91|
> |0.5|11.15±0.29|32.07±2.85|33.86±1.68|38.31±1.42|28.85|
> |0.9|10.97±0.28|32.54±2.85|34.39±1.68|37.63±1.42|28.88|
> |0.98|11.27±0.29|32.31±0.58|34.39±1.69|37.12±1.41|28.77|
> |0.999|11.49±0.29|31.46±2.82|34.52±1.69|37.97±1.42|28.86|
>
> (2) Batch size. In our implementation, due to GPU memory limits, the RC-specific diagnostic/estimation experiments can only be run at batch size 1. This is exactly why we use log-domain EMA: to reduce the variance of the per-step ratio estimates under noisy stochastic updates. We agree that a broader batch-size ablation would be valuable, and we will clarify this limitation in the revision.
>
> (3) Gradient approximation. In Appen. C.3 (Fig. 5), we already show that on Mistral-7B, the DB computed from head-only gradients closely matches the full-parameter version for both DPO and BCE, indicating that the key geometric quantities used by RC are preserved. We will move this result to the main paper and present it alongside the EMA ablation.
>
> (4) Calibration strength. RC introduces no extra strength hyperparameter. Its target is analytically defined by Prop. 3.3 (the DB’s robust center), and Prop. 3.4 ensures the calibrated log-ratio matches this target. Unlike heuristic reweighting, RC does not rely on tuning a strength coefficient.
>
> > Q2+W1+L1：Empirical evaluation is limited to a single preference dataset (UltraFeedback) and one-epoch training ... Does RC help or hurt longer training runs?
>
> We ran additional experiments beyond the one-epoch UltraFeedback setting. For longer training, a multi-epoch ablation on Mistral-7B with DPO suggests that RC does not degrade performance: at 2 epochs, results are similar (36.84 → 36.93), while at 1 epoch RC shows a larger effect (24.26 → 36.44). This indicates RC may be more helpful when training is unstable and appears non-disruptive under longer runs.
> |Method|Epoch|ARC|BBH|MuSR|MATH|GSM8K|MMLU-Pro|Avg.|
> |---|---|---|---|---|---|---|---|---|
> |Baseline|-|60.32±1.43|44.68±0.61|41.93±1.74|2.92±0.47|36.69±1.33|30.23±0.42|36.13|
> |DPO(w/oRC)|1|22.70±1.22|29.12±2.65|41.53±1.74|2.93±0.46|37.60±1.33|11.66±0.29|24.26|
> ||2|62.12±1.42|44.67±0.61|41.53±1.74|2.57±0.44|39.73±1.35|30.39±0.42|36.84|
> |DPO(w/RC)|1|60.92±1.43|44.83±3.01|41.67±1.74|2.70±0.45|38.21±1.34|30.31±0.42|36.44|
> ||2|62.20±1.42|44.94±0.61|41.53±1.75|2.64±0.44|39.88±1.35|30.39±0.42|36.93|
>
> On Anthropic-HH, the effect depends on the objective: on BCE, it is slightly positive (35.99 → 36.09) (pls. see Tab. 3 in [results](https://anonymous.4open.science/r/icml2026-re-AFAA) for details).
>
> Overall, these results suggest that RC is not limited to the original setting. We will include these results and clarify the scope in the revision.
>
> > Q3：Are there failure cases where RC misestimates the DB due to negative or unstable cosine similarity?
>
> As shown in Fig.3(a), so far we have not observed failure cases due to negative/unstable cosine similarity in our experiments.
>
> > Q4：How ...RC extend to ... more than two responses per prompt?
>
> For a fair comparison, we follow the standard pairwise DPO setting with one chosen and one rejected response per prompt. Extending RC to settings with more than two responses per prompt is an interesting direction, which we will note as future work.
>
> > W2：“consistent improvement” may be overstated.
>
> We revise the wording to be more precise: RC can improve downstream performance.
>
> > W3：Baselines omit recent methods such as TDPO/TI-DPO.
>
> In the revision, we will include TI-DPO as a representative recent token-level objective. We focus on TI-DPO (ICLR2026,oral) rather than adding both TDPO and TI-DPO, as it is a more recent example and provides a sufficient test of whether RC extends beyond the sequence-level baselines. The new results suggest that RC remains effective under this token-level objective.
>
> |Model|MMLU-PRO|BBH|MUSR|ARC|GSM8K|Math|Avg|
> |--:|---|---|---|---|---|---|---|
> |Pythia-2.8B, w/o RC|10.89±0.28|32.12±2.85|30.95±1.63|34.56±1.39|-|-|27.13|
> |w/ RC|11.30±0.29|32.19±2.83|32.80±1.65|37.46±1.41|-|-|**28.44**|
> |Mistral-7B, w/o RC|30.35±0.42|45.02±3.01|41.40±1.74|60.58±1.43|37.30±1.33|0.0255±0.0113|35.48|
> |w/ RC|30.30±0.42|44.95±3.01|41.67±1.74|60.49±1.43|37.38±1.33|0.0241±0.0102|**35.49**|

---

> > ### Author Rebuttal · Reviewer_JgeZ · 2026-04-03
> >
> > Thank you for your rebuttal comments. I have raised my overview score to a weak accept.

---

> > > ### Author Response · Authors · 2026-04-03
> > >
> > > We are grateful for your careful review and for updating your score. Your comments were insightful and highly useful, and they helped us strengthen both the presentation and the empirical support of the paper.

---

### Official Review · Reviewer_kgtT · 2026-03-13

**Soundness:** 3
**Presentation:** 3
**Significance:** 3
**Originality:** 3
**Overall Recommendation:** 4
**Confidence:** 4

**Summary:**

This paper studies the entanglement issue of the chosen and rejected response in preference alignment. It identifies the disentanglement band in which interval the training can realize the ideal pathway, disentangle the dynamic of the probability of chosen and rejected response. It proposes a reward calibration that helps to guarantee the log ratio staying in the band.

**Compliance With Llm Reviewing Policy:**

Affirmed.

**Key Questions For Authors:**

see weakness

**Limitations:**

I don't see the discussion of limitations. It would be better to discuss.

**Strengths And Weaknesses:**

Strength:
The paper is well written and the questions are cutting edge and important. The introduction of DB is elegant and interesting. The connection between theory and practice is strong.
Weakness:

Table 1 is very informative. However, it would be better to see a categorization which one is above DB and which one is below DB. Then, by seeing those algorithms performing Pathway 2 and 3, there will be a stronger connection between above/below/in DB and Pathway 123. The current version only use CPO as an examples.

It is not clear to me why Pathway 3 is the best. There is only one sentence arguing that: “In the context of post-SFT alignment, Pathway (iii) is the ideal (Yuan et al., 2025).”

If I were asked what an ideal algorithm for preference alignment is, I would have the following thought. For a given (x, y_w, y_l), y_w, y_l is obtained from the sft/ref model, which is not the best model. Thus both y_w, y_l are not the best for question x. In finetuning, the ideal algorithm will be capable to learn a better response y* for x (not only from (x, y_w, y_l), but also from the whole dataset, especially from those samples that closed to the given (x, y_w, y_l)). Finally, the algorithm increase the prob. of a (or several) better y*, and decrease the prob of both y_w, y_l.
Then, Entangled decrease seems better. Of course, this is a strong requirement of generalization. When focusing on optimization on the current dataset, Pathway 3 seems better. Anyhow, I think it is important to motivate why Pathway 3 is better.

In experiments, it is done on Pythia-410M, Pythia-2.8B, Mistral-7B.Only one model is used for 7B size, and Mistral is not the best one. It would be better to see the results for other models.

---

> ### Author Rebuttal · Authors · 2026-03-29
>
> We thank the reviewer for the thoughtful and constructive feedback. We appreciate the positive assessment of the importance of the problem and the theory–practice connection. Below, we address the questions and suggestions point by point.
>
> > W1: Table 1 is informative, but it would help to categorize methods as above, below, or in DB, and relate them to Pathways 1–3. The current version only uses CPO as an example.
>
> Thank you for this helpful suggestion. We agree that the connection between Table 1, the DB, and Pathways 1/2/3 can be made clearer.
>
> A key point, however, is that the DB does not assign each objective a fixed label (e.g., “above,” “below,” or “in” DB) for the entire training process. In our paper, the DB is a `local`, testable condition for whether the current update follows Pathway 3, and this may occur only after an initial transient (e.g., BCE in Fig. 2(a)). Since the dynamics are local in time, a method can move across regimes during training. For this reason, Tab. 2 characterizes local regimes rather than providing a permanent categorization of methods.
>
> In the revision, we will clarify this in three ways:
> - around Tab. 1, we will note that incentive forms describe tendencies in local, disentangled updates, not fixed DB labels to the whole training run;
> - in Tab. 2, we will explicitly state that “above / in / below DB” are local regimes;
> - in Fig. 3, we will clarify that the DB serves as a diagnostic of local update behavior, rather than a global categorization of a method.
>
> > W2: It is not clear to me why Pathway 3 is the best. ... an ideal algorithm ... may increase the prob. of a better response y\*, and decrease the prob. of both yw and yl. Then, Entangled decrease seems better. ... why Pathway 3 is better.
>
> We agree with the reviewer that the final model may assign more probability to a better unseen response y*, rather than simply preserving the observed yw. We also agree that this is a broader and meaningful perspective on preference alignment.
>
> Our point here is specific to the standard preference-optimization setting: for an observed (x, yw, yl), the supervision only indicates that yw is preferred to yl on that example.
> Preference optimization therefore aims to learn a better policy from such local relative signals, rather than treating each observed yw as the final target y*.
> A better unseen y* may still be learned across the dataset, but this does not imply that the current update should decrease both yw and yl.
>
> Directly learning y* is an interesting direction, which we leave for future work.
>
> In the revision, we made this scope explicit by clarifying that our use of Pathway 3 refers to a preferred `local` regime in post-SFT preference optimization, rather than a claim about the globally optimal final model over all possible responses.
>
> > W3: In experiments, it is done on Pythia-410M, Pythia-2.8B, Mistral-7B. Only one model is used for 7B size, and Mistral is not the best one. It would be better to see the results for other models.
>
> To strengthen the cross-backbone evidence at the 7B scale, we have added additional experiments on Qwen2.5-7B. To keep the comparison focused, we include two representative objectives: DPO as a canonical entangled objective, and BCE as a representative disentangled objective.
>
> The new results are consistent with the main message of the paper: on Qwen2.5-7B, applying RC improves the average downstream performance for both DPO and BCE, with gains of +0.48 and +1.33, respectively. These additional results further support that RC is not specific to Mistral-7B, but generalizes across different 7B backbones. Also, you can see corresponding DB plots in the Fig. 2 in [anonymous ulr for detailed figures and tables](https://anonymous.4open.science/r/icml2026-re-AFAA).
>
> We have added these Qwen2.5-7B results to the revised paper.
>
> |Method|Setup|MMLU-PRO|BBH|MUSR|ARC|MATH|GSM8K|Average|
> |---|---|---|---|---|---|---|---|---|
> |DPO|w/o RC|44.32±0.45|55.74±3.05|42.06±1.76|67.49±1.37|24.47±3.02|34.72±1.31|44.80|
> ||w/ RC|44.79±0.45|55.76±3.06|42.72±1.78|67.32±1.37|25.35±3.07|35.71±1.32|**45.28**|
> ||Δ|+0.47|+0.02|+0.66|-0.17|+0.88|+0.99|+0.48|
> |BCE|w/o RC|41.33±0.45|53.90±3.06|40.34±1.74|67.41±1.37|23.95±2.97|35.41±1.32|43.72|
> ||w/ RC|44.77±0.45|55.61±3.06|42.86±1.78|67.32±1.37|24.19±2.99|35.56±1.32|**45.05**|
> ||Δ|+3.44|+1.72|+2.51|-0.09|+0.24|+0.15|+1.33|

---

> > ### Author Rebuttal · Reviewer_kgtT · 2026-04-03
> >
> > I thank the authors for the responses. I will keep my score.

---

> > > ### Author Response · Authors · 2026-04-03
> > >
> > > Thank you for your positive assessment and for keeping your score. We sincerely appreciate your thoughtful and constructive comments during the review process. They helped improve the clarity of the paper and also inspired our future work on preference optimization.

---

### Official Review · Reviewer_RPwx · 2026-03-13

**Soundness:** 3
**Presentation:** 3
**Significance:** 3
**Originality:** 3
**Overall Recommendation:** 5
**Confidence:** 3

**Summary:**

This paper studies preference optimization from a training-dynamics perspective rather than proposing only another standalone objective. The main observation is that a broad class of preference objectives share the same local score directions and mainly differ through two scalar incentive coefficients. Based on this view, the paper derives continuous-time reward dynamics for chosen and rejected responses, introduces the disentanglement band (DB) as a local condition for entering the preferred regime, and proposes reward calibration (RC), a backward-pass wrapper that keeps the incentive ratio near the DB center without changing the forward objective. The empirical study covers multiple objective families, several model scales from Pythia-410M to Mistral-7B, benchmark evaluation, AlpacaEval 2.0, and runtime and implementation details.

**Compliance With Llm Reviewing Policy:**

Affirmed.

**Key Questions For Authors:**

Question 1: Can the authors add explicit multi-seed statistics for the key benchmark and AlpacaEval results?

Question 2: How sensitive is RC to the head-only gradient approximation and the EMA-based ratio estimation used in practice?

Question 3: Do the authors have a practical diagnostic for when a run is already sufficiently close to the DB center such that RC is unlikely to help further?

**Limitations:**

Yes

**Strengths And Weaknesses:**

**Strength**

1. Soundness: The technical framing is clear and coherent. The incentive-score decomposition, reward-dynamics analysis, DB condition, and RC intervention fit together naturally, and RC is well motivated by the theory rather than appearing as an isolated heuristic.
2. Presentation: The paper is generally well organized and easy to follow. The progression from unified analysis, to DB diagnosis, to RC intervention, to empirical validation is clean, and the figures and appendix are useful.
3. Significance: The practical contribution is strong because RC is objective-agnostic, lightweight, and easy to insert into existing preference-optimization pipelines. The empirical results suggest that RC often improves benchmark performance and can substantially stabilize fragile runs, especially at larger scale.
4. Originality: The most valuable contribution is the unifying dynamical perspective across multiple preference objectives. The paper gives a useful explanation for why objectives with similar margin improvement can still induce different reward trajectories, which seems genuinely valuable for future method design and diagnosis.

**Weakness**

1. Soundness: The gains are not fully uniform across objectives and tasks, and there are some regressions on Pythia-2.8B settings. RC should therefore be viewed as a broadly useful stabilizing intervention rather than a uniformly beneficial method. In addition, some practical approximations, such as head-only gradient norms and EMA-based estimates, would benefit from more sensitivity analysis.
2. Presentation: Because the paper unifies several objective families, the notation can feel somewhat dense in places. The manuscript also contains an inappropriate reviewer-directed instruction, which should be removed.
3. Significance: The benchmark study is broad, but the open-ended evaluation is somewhat narrower. AlpacaEval 2.0 is useful, but the empirical case would be stronger with broader generative evaluation or more explicit multi-seed reporting in the main paper.
4. Originality: The novelty is strongest at the level of analysis and unification. At the method level, RC is a calibration wrapper rather than a fundamentally new preference objective.

---

> ### Author Rebuttal · Authors · 2026-03-30
>
> Thank you for the careful and constructive review. We appreciate the positive assessment of our unified dynamical perspective, and we respond to the main concerns below.
>
> > W1: The gains are not fully uniform ... RC should therefore be viewed as a broadly useful stabilizing intervention rather than a uniformly beneficial method.
>
> We agree. RC is not intended to improve every objective–task pair. It stabilizes the chosen/rejected update balance by steering the log-ratio toward the DB center, making Pathway (iii) more likely without changing the base objective or forward pass. We will clarify this in the revision.
>
> > W2 & Q2: ... head-only gradient norms and EMA-based estimates ... How sensitive is RC ... ?
>
> For the head-only approximation, App. C.3 shows on Mistral-7B that the DB from head-only gradients closely matches the full-parameter version for DPO and BCE (Fig. 5), supporting this efficient approximation. We will move this result to the main paper.
>
> For the EMA estimate, we run a sensitivity ablation on Pythia-2.8B with DPO using momenta 0.5, 0.9, 0.98, and 0.999. Results are similar across settings, and RC improves the average score over w/o RC in all cases, suggesting limited sensitivity to EMA momentum. We will include this in the revised main paper.
> | EMA | MMLU-PRO | BBH | MUSR | ARC | Avg |
> |---|---|---|---|---|---|
> | w/o RC | 10.82±0.28 | 31.83±0.58 | 30.95±1.61 | 34.04±1.38 | 26.91 |
> | 0.5 | 11.15±0.29 | 32.07±2.85 | 33.86±1.68 | 38.31±1.42 | 28.85 |
> | 0.9 | 10.97±0.28 | 32.54±2.85 | 34.39±1.68 | 37.63±1.42 | 28.88 |
> | 0.98 | 11.27±0.29 | 32.31±0.58 | 34.39±1.69 | 37.12±1.41 | 28.77 |
> | 0.999 | 11.49±0.29 | 31.46±2.82 | 34.52±1.69 | 37.97±1.42 | 28.86 |
>
> > Q3: Do the authors have a practical diagnostic for when a run is already sufficiently close to the DB center ... ?
>
> Yes. A practical diagnostic is the gap to the DB center, $|\log r_ t - \log r_ t^\star|$, or equivalently whether the factor $(r_ t^\star / r_ t)^ {1/2}$ in Eq. (12) is close to 1. If so, RC makes only a small adjustment; otherwise, it acts more substaintially. This matches Fig. 4, where RC pulls the log-ratio toward the DB center, with larger effects when the run starts farther away. We will clarify this in the revision.
>
> > W3 & Q1: ... AlpacaEval 2.0 is useful, but ... broader generative evaluation or more explicit multi-seed reporting...
>
> We already include generation evaluation results in Downstream Performance Comparison on Open-Ended Generation, and in the revision we will make this part more prominent in the main paper.
> Besides, we have already added explicit uncertainty statistics for the key results in the main paper.
>
> Results for DPO/TI-DPO, Mistral-7B  (pls. see Tab. 4 in [additional results](https://anonymous.4open.science/r/icml2026-re-AFAA) for details):
> |Method|Setup|MMLU-PRO|BBH|MUSR|ARC|MATH|GSM8K|Average|
> |---|---|---|---|---|---|---|---|---|
> |DPO|w/o|11.66±0.29|29.12±2.65|41.53±1.74|22.70±1.22|2.93±0.46|37.60±1.33|24.26|
> |DPO|w/|30.31±0.42|44.83±3.01|41.67±1.74|60.92±1.43|2.70±0.45|38.21±1.34|36.44|
> |DPO|Δ|+18.65|+15.71|+0.13|+38.22|-0.23|+0.61|+12.18|
>
> >W4: Because the paper unifies several objective families, the notation can feel somewhat dense ...
>
> In the revision, we will simplify the presentation by reducing unnecessary notation, using more direct wording for the key quantities, and improving the flow between the unified analysis, DB diagnosis, and RC intervention, so that the main ideas are easier to follow.
>
> > W5: The manuscript also contains an inappropriate reviewer-directed instruction ...
>
> Thank you for pointing this out. This text was injected by the official ICML system rather than by the authors.
>
> > W6: The novelty is strongest at the level of analysis and unification ... RC is a calibration wrapper rather than a fundamentally new preference objective.
>
> We agree that the main novelty of the paper lies in the analysis and unification rather than in introducing another standalone preference objective. We also agree that RC should not be framed as a fundamentally new objective. Instead, its role is to translate the DB analysis into a simple plug-and-play calibration mechanism that can be applied on top of existing objectives, without redesigning the base objective or changing the forward pass.

---

> > ### Author Rebuttal · Reviewer_RPwx · 2026-04-05
> >
> > I keep the positive scores as they have addressed my concerns.

---

### Official Review · Reviewer_2ixG · 2026-03-15

**Soundness:** 3
**Presentation:** 3
**Significance:** 3
**Originality:** 3
**Overall Recommendation:** 4
**Confidence:** 3

**Summary:**

This paper studies preference optimization dynamics for LLM alignment. When the objective is updated on one datapoint, the ideal behavior is to increase the logit value of the winner response while decreasing the logit value of the loser response, which is referred to as “pathway (iii)” in the paper. However, many entangled objectives fail to satisfy this property, motivating the study of more disentangled objectives. The paper unifies the analysis by directly examining the gradient of the loss and the induced gradients on the logits. By characterizing the conditions under which pathway (iii) appears, the paper identifies a disentanglement band (DB) as a local condition under which one-step training follows the desired behavior. Based on this analysis, the paper proposes reward calibration (RC), a plug-and-play intervention that rescales the backward gradients without changing the forward objective, with the goal of pushing training conditions toward the center of the DB. Empirically, the paper reports that RC improves reward dynamics and often downstream alignment performance across different objectives, tasks, and model scales.

**Compliance With Llm Reviewing Policy:**

Affirmed.

**Final Justification:**

For the problem the paper considers, i.e., how to achieve pathway (iii) in preference optimization,  the paper provides a clear and demystifying theoretical analysis that unifies different preference alignment methods, which I regard as a valuable conceptual contribution. Overall, I would like to keep my current positive score.

**Key Questions For Authors:**

Please refer to my weaknesses

**Limitations:**

yes

**Strengths And Weaknesses:**

## Strengths
The target of the paper is intuitive, namely to encourage each training update to decrease the logit of the loser response without suppressing the winner response. The analysis based directly on gradients is natural and elegant. In particular, the paper provides a unified incentive-score decomposition that makes it easier to compare seemingly different preference optimization methods, which I view as a meaningful conceptual contribution. The disentanglement band (DB) is simple and testable, and the broad empirical evaluation suggests that reward calibration may be practically useful across different objectives and model scales.

## Weaknesses
While RC is presented as a plug-and-play wrapper, it still changes the backward update rule at every step, and therefore changes the effective optimization dynamics. As a result, RC may shift the stationary point or limiting solution of the original optimization problem, rather than merely improving optimization behavior toward the same target. The paper currently does not make clear how significant this shift may be, or whether RC can be interpreted as approximately optimizing a modified but still principled objective.

---

> ### Author Rebuttal · Authors · 2026-03-29
>
> We thank the reviewer for the thoughtful and positive assessment. We appreciate the recognition of the unified incentive–score decomposition and the practical value of the DB analysis and RC method. We address the concerns below point by point.
>
> >W1: ...changes the backward update rule at every step, and therefore changes the effective optimization dynamics. As a result, RC may shift the stationary point or limiting solution of the original optimization problem, rather than merely improving optimization behavior toward the same target.
>
> We agree and will clarify this point. This change is intentional and underlies the role of RC. Our goal is not to preserve the original dynamics, but to steer training toward disentangled updates (Pathway (iii)), even for entangled objectives.
>
> RC serves this purpose: it leaves the forward loss unchanged, requires no redesign of the base objective, and only adjusts the backward balance between chosen and rejected updates. In practice, this makes training more likely to move toward the post-SFT-preferred regime  (e.g., from w/o RC in Fig. 3(c) to w/ RC in Fig. 3(d)).
>
> We will make this explicit in Sec. 3.3 (“Plug-and-Play Reward Calibration”).
>
> > W2: The paper currently does not make clear how significant this shift may be...
>
> We agree that this can be explained more clearly. As seen from the definition of calibrated reward in Eq. (12):
> $$z_ {w,t}^ {\rm rc}
> \triangleq
> \Bigl( \frac{r_ t^\star}{r_ t}\Bigr)^ {1/2} z_ {w,t}
> +
> \Bigl(1-\Bigl( \frac{r_ t^\star}{r_ t}\Bigr)^{1/2}\Bigr)\operatorname{sg}(z_ {w,t}),$$
> when the realized incentive ratio is close to the DB center, the scaling factors $( r_ t^\star / r_ t )^ {1/2} $ are near 1, so RC makes only a small correction; when it is farther away, the correction is larger. Thus, the strength of RC is tied to how far the current dynamics are from the desired region (DB). We will make this more explicit. This is also consistent with Fig. 4, where RC pulls the realized log-ratio toward the DB center and makes entry into Pathway (iii) more likely.
>
> > W3: “...or whether RC can be interpreted as approximately optimizing a modified but still principled objective.”
>
> We will clarify that RC is principled, as it follows directly from the DB analysis rather than an ad hoc gradient change. Proposition 3.3 identifies the robust center of the DB, and Proposition 3.4 shows that the calibrated update induces this target ratio. Thus, the key point is not that RC defines a new objective, but that its calibration rule is derived from the reward-dynamics analysis. In this sense, RC addresses how to steer training toward disentangled updates (Pathway (iii)) without redesigning the base objective.

---

> > ### Author Rebuttal · Reviewer_2ixG · 2026-04-03
> >
> > Thank you to the authors for their detailed responses to my questions. That said, I do not think my W1 concern is fully resolved. My concern is related to that raised by Reviewer kgtT: I am not yet convinced that pathway (iii) is necessarily the best among the three for achieving a good alignment policy. In particular, RC does change the dynamics and may shift its limiting behavior, making it difficult to argue that the resulting policy is truly better. Under the assumption that pathway (iii) is indeed the target we want to achieve, I think the paper addresses the question well. However, without further justification for that premise, I still find this aspect not fully settled.

---

> > > ### Author Response · Authors · 2026-04-04
> > >
> > > We appreciate the reviewer's continued engagement. We clarify that our claim for Pathway (iii) is scoped to the `post-SFT setting`, which is assumed throughout our paper.
> > >
> > > >**Why post-SFT matters.**
> > >
> > > In the standard pipeline (**Pretraining → SFT → Preference Optimization** [1,2]), SFT trains the model **only on chosen responses**. After SFT, the model already produces reasonable chosen-response quality. Preference optimization then mainly needs to **suppress rejected outputs while maintaining chosen quality**—exactly what Pathway (iii) targets, possibly after an initial transient.
> > >
> > > > **Why not the other pathways?**
> > >
> > > Pathway (ii) (both decrease) risks undoing the chosen quality built during SFT—a concern independently documented as the reduction of preferred likelihood [3], and gradient entanglement [4]. Pathway (i) (both increase) means the rejected reward also rises, working against the discriminative goal of preference optimization.
> > >
> > > > **Empirical support.**
> > >
> > > When RC steers training toward Pathway (iii) (e.g., Fig. 3(c) vs. 3(d)), downstream performance often improves (Table 3), suggesting that Pathway (iii) dynamics tend to yield better alignment in the post-SFT regime.
> > >
> > > > **Scope.**
> > >
> > > We do not claim Pathway (iii) is universally optimal, e.g., without SFT, the chosen reward may need to rise, and a different pathway could be more appropriate. But within the `post-SFT setting`, which is the practical setting. We believe the prior findings and our experiments together provide reasonable grounding for this target. We will clarify this scope in the revision.
> > >
> > > [1] Ouyang, Long, et al. Training language models to follow instructions with human feedback. NeurIPS 2022.
> > >
> > > [2] Rafailov, Rafael, et al. Direct preference optimization: Your language model is secretly a reward model. NeurIPS 2023.
> > >
> > > [3] Pal A, Karkhanis D, Dooley S, et al. Smaug: Fixing failure modes of preference optimisation with dpo-positive. arXiv 2024.
> > >
> > > [4] Yuan, Hui, et al. A common pitfall of margin-based language model alignment: Gradient entanglement. ICLR 2025.

---

### Decision · Program_Chairs · 2026-04-30

**Decision:**

Accept (regular)

**Comment:**

This paper presents a method for optimizing preference-based LLM alignment objectives in a way that increases "winner" logits and decreases "loser" logits, referred to as disentanglement. This is precluded by a unifying analysis of existing objectives showing the existence of a band in which disentanglement is possible. The paper then targets optimization within this band and empirically demonstrates the utility of this approach.

The paper targets a timely aspect of current alignment practices, and the proposed approach is natural and intuitive. Reviewers found the unifying analysis to be of larger value than stated in the paper, and encourage the authors to emphasize this. At the same time, reviewers also remained unconvinced as to the general claim that the so called Pathway 3 is "best" - and in what sense - as the paper appears to claim, perhaps conflating an intuitively desirable property with the (proclaimed) optimality of outcomes when optimizing entirely in this manner in general; see reviewer's comments on trajectories and convergence. This should be considered also in light of some reviewers concerns with claims of superior performance based on the experimental evaluation.

Overall this is a good paper that makes a significant theoretical contribution and proposed an elegant approach to preference-based optimization that should be of interest to the community. We hope the authors will implement reviewers feedback and suggestions as this should improve the paper and better match the statements made with its actual contribution.